# CONCEPT BOTTLENECK LANGUAGE MODELS FOR PROTEIN DESIGN

**Aya Abdelsalam Ismail**[1,2]    **Tuomas Oikarinen**[3]    **Amy Wang**[1,2]    **Julius Adebayo**[4]
**Samuel Stanton**[1,2]    **Héctor Corrada Bravo**[1]    **Kyunghyun Cho**[1,2,5,6]    **Nathan C. Frey**[1,2]
[1]Genentech    [2]Prescient Design    [3]University of California San Diego    [4]Guide Labs
[5]Department of Computer Science, New York University
[6]Center for Data Science, New York University

## ABSTRACT

We introduce Concept Bottleneck Protein Language Models (CB-pLM), a generative masked language model with a layer where each neuron corresponds to an interpretable concept. Our architecture offers three key benefits: i) Control: We can intervene on concept values to precisely control the properties of generated proteins, achieving a $3\times$ larger change in desired concept values compared to baselines. ii) Interpretability: A linear mapping between concept values and predicted tokens allows transparent analysis of the model's decision-making process. iii) Debugging: This transparency facilitates easy debugging of trained models. Our models achieve pre-training perplexity and downstream task performance comparable to traditional masked protein language models, demonstrating that interpretability does not compromise performance. While adaptable to any language model, we focus on masked protein language models due to their importance in drug discovery and the ability to validate our model's capabilities through real-world experiments and expert knowledge. We scale our CB-pLM from 24 million to 3 billion parameters, making them the largest Concept Bottleneck Models trained and the first capable of generative language modeling.

## 1 INTRODUCTION

Protein Language Models (pLMs) have emerged as a prominent framework for protein representation learning, encapsulating millions of years of protein evolution (Hayes et al., 2024). These models have demonstrated exceptional performance on intricate tasks such as predicting protein structure and function (Lin et al., 2023; Chen et al., 2023; Elnaggar et al., 2022; Xu et al., 2023). Furthermore, they have opened new frontiers in diverse and critical applications, including healthcare and drug discovery (Hie et al., 2024; Hayes et al., 2024). Despite their impressive capabilities, controlling, interpreting, and debugging pLMs remains challenging due to the opaque nature of transformers (Vaswani, 2017).

Recently, some pLMs have supported conditional generation by adding various protein input representations, functions, and properties as *tags* (which can be viewed as concepts) and predicting them during training (Madani et al., 2020; Hayes et al., 2024; Ruffolo et al., 2024). However, these models can selectively rely on or ignore different aspects of the input. Even if these models offer some form of control, there is no effective way to interpret or debug them. Domain experts might want to know *which concept the model depends on the most to generate the amino acid "E"*, *which amino acids are correlated with enzymatic activity*, or answer counterfactuals like *if we want to decrease hydrophobicity, which amino acid should we replace "E" with*, which is not possible with existing models. Debugging language models is notoriously difficult, and it is often unclear what a model has learned or when it will fail. These limitations lead to distrust among domain experts, as humans tend to **not trust what they do not understand**.

Our work relies on the fact that neural networks *do not have to be black boxes*; since we construct and train these networks, we can design and align them according to our needs. This is a unique advantage of building models from scratch, with control, interpretability, and debugging in mind from the outset (Frey et al., 2024; Adebayo et al., 2023). To achieve explainability and alignment, it

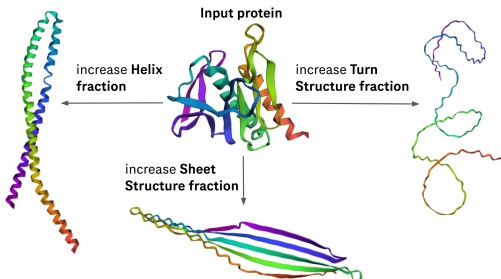

Figure 1: Changing protein structure using **C**oncept **B**ottleneck **P**rotein **L**anguage **M**odel.

is essential to use intermediate representations based on well-understood concepts. To this end, we present *Concept Bottleneck Language Models*, a novel architecture for large language models that relies on human-understandable concepts via a concept bottleneck layer and auxiliary loss terms. This architecture enables concept-controllable generation and explanations, compelling the model to align with human-understandable concepts and providing the means to debug the model and understand its limitations. Our approach builds on the foundation of concept bottleneck generative models (CBGMs) (Ismail et al., 2023).

In this paper, we focus on protein design, a domain where control (Gruver et al., 2024) and interpretability (Adebayo et al., 2023) are critically needed. Protein design benefits from a set of well-defined biophysical and bioinformatics properties that domain experts understand and wish to control. We introduce *Concept Bottleneck Protein Language Model* (CB-pLM). We summarize our contributions as follows:

- We propose a novel architecture, training, and intervention schemes for large language models that incorporate human-understandable concepts. This results in a controllable, interpretable, and debuggable LLM by design. We focus on masked protein language models, given their critical role in high-stakes applications such as drug development, where the need for control and interpretability is paramount. We train masked protein language models with 24M, 150M, 650M, and 3B parameters using the proposed CB-pLM architecture with over 700 concepts. We do not observe a significant increase in perplexity compared to unconstrained models.
- We compare CB-pLM to various conditional pLMs of the same size, all trained on the same dataset, across more than 80 single and multi-property control *in silico* experiments. Our model demonstrates $3\times$ better control in terms of change in concept magnitude and a $16\%$ improvement in control accuracy compared to other architectures. Additionally, we benchmark CB-pLM against state-of-the-art (SOTA) protein design models explicitly trained to optimize a single concept. Remarkably, our general-purpose CB-pLM, trained to learn over 700 concepts, delivers results comparable to SOTA models, while maintaining the naturalness of the protein.
- We demonstrate how CB-pLM allows domain experts to assess what the model has learned, i.e., *interpretability*, and identify and correct any unwanted behavior in the model, i.e., *debuggability* by simply visualizing the weights from the model's decoder layer.

## 2 CONCEPT BOTTLENECK LANGUAGE MODEL

We aim to create a language model that we can easily control, interpret, and debug. To achieve this, we modify the standard masked language model architecture by adding a concept bottleneck layer to explicitly incorporate human-understandable concepts into the language model—termed a concept-bottleneck (masked) language model (CB-LM). The concept layer can then be used to **control, interpret, and debug** the model. This section discusses the proposed language model architecture, loss functions, training and intervention procedure.

**Setup** Each sequence $x$ is represented with a set of tokens, i.e., $x = [x_0, ..., x_s]$, where $s$ is the maximum sequence length. Each sequence starts with the CLS token i.e. $x_0 = $ `[CLS]` followed by a sequence of amino acids ending with `[EOS]` as well as padding tokens for sequences below maximum length. Additionally, each sequence is annotated with a set of *global* predefined human-understandable concepts $c = [c_0, ..., c_k]$ where $k$ is the number of concepts. Taken together, this results in a training dataset consisting of token-concept pairs $\{(x, c)_i\}_{i=1}^n$ where $n$ is the number of samples.

## 2.1 ARCHITECTURE

The overall architecture is shown in Figure 2. We start with a standard BERT-based masked language model (MLM) (Devlin et al., 2018), which is commonly used in many protein language models (Rives et al., 2019; Lin et al., 2022; Hayes et al., 2024; Frey et al., 2024). A mask token `[MASK]` is randomly applied to a subset of tokens in the sequence $x$ resulting in a masked version of the sequence. After masking, $x$ is passed into a multi-head self-attention transformer encoder, and the output of the encoder is denoted as $\mathbf{H} = \text{TransformerEncoder}(x)$,

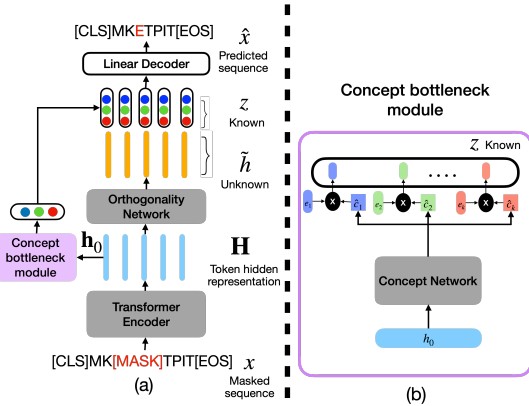

Figure 2: Concept Bottleneck Language Model

where $\mathbf{H} = [\mathbf{h}_0, \ldots, \mathbf{h}_s]$. We introduce three main architectural changes in CB-LM: (a) concept bottleneck module (CB-module), (b) orthogonality network, and (c) linear decoder.

**Concept bottleneck module** After $x$ passes through the encoder, the hidden representation of the classification `[CLS]` token, $\mathbf{h}_0$, which serves as an aggregate representation of the sequence, is passed to the CB-module shown in Figure 2b. The concept network (Koh et al., 2020) learns a function $g$ that maps the input sequence into concept predictions $\hat{c} = g(\mathbf{h}_0)$. Each concept is also represented with a learnable embedding $e$. The final representation of concept $i$ is $z_i = \hat{c}_i \times e_i$. The output of the CB-module, $z = [z_1, z_2, \ldots, z_k]$, is referred to as the *known* embedding, as we know exactly which concept each neuron in the embedding corresponds to.

**Orthogonality network** This network aims to produce a token representation devoid of any concept information, which we refer to as the *unknown* token embedding $\widetilde{h}$. This is achieved by enforcing an orthogonality loss, as described in Section 2.2, between its output and the known concepts.

**Linear decoder** The known and unknown embedding are concatenated and passed into a single linear layer followed by a softmax function, i.e., $\hat{x}_{\text{M}} = \text{softmax}(W^{\text{dec}} \cdot [z, \widetilde{h}_{\text{M}}])$, where $x_{\text{M}}$ represents a masked token `[MASK]`. We use a linear layer as the decoder for interpretability and debugging purposes. A linear layer allows us to calculate the contribution of each concept to the final prediction by simply multiplying the concept activation value with its weights. This gives us an interpretable answer to questions such as *Which concept does the model depend on the most to generate the amino acid"E"?* or counterfactual questions like *if we intervened on the concept strength of hydrophobicity, which amino acid would the decoder output?*. Debugging neural networks is notoriously difficult; however, our setup simplifies this process. Since we have prior knowledge of how challenging a concept is to learn, we can identify and fix bugs by inspecting the concept weights.

Examining the weight matrix $W^{dec}$ can (a) help us understand the correlation between different concepts, aiding in the detection of any spurious correlations learned by the model, (b) elucidate the correlation between concepts and tokens, answering questions like *which amino acids are correlated with enzymatic activity?*, and (c) identify concepts that we can control and those we cannot, i.e., if the weight for a concept is small, then the model will not be able to control that particular concept.

## 2.2 LOSS FUNCTIONS AND TRAINING

**Loss functions** Following CBGMs (Ismail et al., 2023), concept bottleneck language models are trained with three losses: a standard masked language modeling loss $\mathcal{L}_{\text{MLM}}$, concept loss $\mathcal{L}_{\text{Concept}}$ and an orthogonality loss $\mathcal{L}_{\text{orth}}$. The final loss is given by $\mathcal{L}_{\text{total}} = \mathcal{L}_{\text{MLM}} + \alpha\mathcal{L}_{\text{Concept}} + \beta\mathcal{L}_{\text{orth}}$, where $\alpha$ and $\beta$ are hyperparameters.

- *Generative masked language modeling loss:* $\mathcal{L}_{\text{MLM}} = -\mathbb{E}_{x,m}\left[\frac{1}{m}\sum_{i \in m} \log P(x_i \mid x_{\backslash m})\right]$, where $m$ is the set of positions of the randomly masked tokens, and the model is trained to predict the identity of the masked tokens.
- *Concept loss:* Given the general-purpose nature of large language models, they are expected to handle thousands of concepts with categorical or real values. Often, samples lack values for most concepts. To address this, we normalize all real-valued concepts to [0, 1] and apply mean square

error loss on the entire concept embedding: $\mathcal{L}_{\text{Concept}} = \frac{1}{k} \sum_{i=1}^{k} (c_i - \hat{c}_i)^2$. Missing values are replaced with default values, and their effect is removed from the loss function by considering only non-missing concepts, achieved by masking the errors before backpropagation.

- *Orthogonality loss:* Similar to Ismail et al. (2023), we encourage the known and unknown embeddings to encode different information by applying an orthogonality constraint (Ranasinghe et al., 2021), minimizing the cosine similarity between them. This loss encourages, but does not guarantee, disentanglement.

**Training** *CBM:* We used independent training for the CB-layer (Koh et al., 2020) to reduce concept leakage (Mahinpei et al., 2021). *Token regularization:* Feature attribution is needed for coordinate selection during intervention, as discussed in section 2.3. To ensure attribution is reliable, we followed Adebayo et al. (2023)'s recommendation and added Gaussian noise to the token embedding during training. Additional details are available in Appendix A.2.

## 2.3 Coordinate selection and Interventions

CB-LM enables concept-level test-time interventions, allowing for global sequence control while maintaining token-level predictions. During training, tokens are randomly masked and predicted, which is not ideal for interventions. Instead, we wish to identify tokens that most significantly affect a concept and intervene on those (Gruver et al., 2024; Adebayo et al., 2023). For example, to increase the Aromaticity of a protein, we target amino acids that decrease Aromaticity. This process is called *coordinate selection*.

A straightforward method to measure a token's effect on a concept is occlusion, where each token is replaced with a reference value (e.g., [MASK]) one at a time to observe changes in the predicted concept. While effective, this method is slow for large-scale applications. Instead, we use a gradient-based approximation of occlusion, requiring only a single forward pass. We propose using a single step of integrated gradients (Sundararajan et al., 2017), which can be seen as a first-order Taylor approximation of occlusion. Since our inputs are discrete, we calculate gradients with respect to the learned token embeddings and sum over the embedding dimension. Let $T \in \mathbb{R}^{v \times d}$ be the learned token embeddings, where $v$ is the vocabulary size and $d$ is the transformer hidden dimension. Then, let $f_T(x) = [T_{x_0}, T_{x_1}, \ldots, T_{x_s}]$. The attribution $A$ for token $t$ of input $x$ on concept $i$ is:

$$A(x, i, t) = (T_{x_t} - T_{\text{[MASK]}})^\top \nabla_{T_{x_t}} \hat{c}_i(f_T(x)),$$

where $T_{\text{[MASK]}}$ is the embedding for the mask token and $\hat{c}_i(f_T(x))$ is the concept value predicted by our CB-Layer for concept $i$.

Informally, $A(x, i, t)$ can be seen as quantifying how the choice of token $x_t$ influences $c_i$ in context of the other tokens $x_{-t}$. If in the training dataset $x_t$ co-occurs with below average values of $c_i$ when combined with $x_{-t}$, then we expect $A(x, i, t)$ to be negative. Similarly if $x_t$ co-occurs with above average values of $c_i$ in the context $x_{-t}$ then we expect $A(x, i, t)$ to be positive. Hence if we want to intervene to *increase* the concept value we greedily choose the top-$k$ coordinates sorted by $A$ in descending order, and if we want to *decrease* the concept value we greedily choose the bottom-$k$ coordinates. In Appendix D.1, we show that this attribution method closely matches the performance of occlusion and improves control effectiveness by $2\times$ over random masking.

## 3 Protein Representation Learning and Design with Conditional Language models

One can view a concept bottleneck language model as a special type of conditional language model. Current conditional large protein language models involve appending conditioning [Tags] to the sequence along with the input (Madani et al., 2020; Shuai et al., 2021; Hayes et al., 2024) and then using these tags to steer the output. To compare the effect of different architectures on protein representation learning, and design, we train three types of conditional models on the same training dataset (sequences and concepts) with comparable sizes.

- **Conditional Protein Language Model (C-pLM)**: These are traditional conditional models, the input to the model is sequences and concepts (which can be viewed as [Tags]); the model is trained to minimize the generative masked language modeling loss, i.e., $\mathcal{L}_{\text{total}} = \mathcal{L}_{\text{MLM}}$.

- **Conditional Protein Language Model with Classifier (CC-pLM)**: This class of models has an extra regression head used to predict the concepts (Hayes et al., 2024). The input to the model are sequences and concepts; the model is trained to minimize the generative masked language modeling loss and concept loss, i.e., $\mathcal{L}_{\text{total}} = \mathcal{L}_{\text{MLM}} + \alpha\mathcal{L}_{\text{Concept}}$.
- **Concept Bottleneck Protein Language Model (CB-pLM) (Ours)**: A family of protein language models with the architecture shown in Figure 2, $\mathcal{L}_{\text{total}} = \mathcal{L}_{\text{MLM}} + \alpha\mathcal{L}_{\text{Concept}} + \beta\mathcal{L}_{\text{orth}}$.

CB-pLM is designed to be controllable, interpretable, and debuggable. Other conditional language models offer control through [Tags]; however, the model can ignore these tags during generation and fail to learn the underlying biophysical concepts for proteins. CB-pLM learns concepts as an intermediate step and includes a loss to encourage disentanglement between concepts and unknown embeddings. Since these concepts are *inside the model*, this forces the model to learn and use these concepts for predictions, leading to better control (Section 4). Furthermore, CB-pLM provides mechanisms for interpretation and debugging (Section 5), which are not available in other conditional models.

**Training Data** We combined sequences from UniRef50 (Suzek et al., 2015) and SWISS-PROT (Bairoch & Apweiler, 2000), removing duplicates. Annotations such as protein clusters, organisms, taxons, biological processes, cellular components, and molecular functions were used as concept annotations. Biopython (Cock et al., 2009) was used to extract biophysical and bioinformatics sequence-level concepts. Overall, all models support 718 concepts; details are in Appendix B.1.

**Architecture and Training** Following Frey et al. (2024), we optimized the architecture and training for efficiency. We removed biases in attention blocks and intermediate layers, increased the effective batch size, used gradient accumulation, and set the masking rate to 25%. We employed the AdamW, mixed precision training, and gradient clipping for stability. More details are in Appendix B.2.

## 3.1 MODEL QUALITY

Model quality is evaluated using perplexity, which measures how well the model predicts a token based on its sequence. Perplexity ranges from 1 (indicating a perfect model) to the vocabulary size (33 for the pLMs vocabulary (Lin et al., 2022)), with higher values indicating more random predictions. To assess performance, we used 10,000 randomly sampled antibodies from the publicly available Mason dataset Mason et al. (2021). Table 1 shows the perplexity of different models on this dataset.

| Models | Tags | Cond. | 24M | 150M | 650M | 3B |
|---|---|---|---|---|---|---|
| LBSTER | No | No | 8.20 | 2.97 | - | - |
| ESM2 | No | No | - | 4.84 | 3.41 | 3.01 |
| C-pLM | Yes | Yes | 4.57 | 2.62 | - | - |
| CC-pLM | Yes | Yes | 4.07 | 2.74 | - | - |
| CB-pLM | No | Yes | 4.29 | 3.20 | 2.48 | 2.50 |

Table 1: Mason dataset perplexity.

LBSTER (Frey et al., 2024) and ESM2 (Lin et al., 2022) are standard masked language models that do not require additional tags during inference and do not support conditioning. Conditioning baselines C-pLM and CC-pLM require ground-truth concepts as inputs during inference; they were given Biopython concepts for sequences, while other concepts were set as missing values. All models learned the training distribution well, with similar-sized models showing comparable perplexity. CB-pLM has a perplexity comparable to conditional models, even though C-pLM and CC-pLM received additional tags as input, while CB-pLM did not. Notably, CB-pLM often achieves higher perplexity than state-of-the-art open-source protein-masked language models of similar size. These results are promising, as CB-pLM, despite its constraints, shows no significant performance drop compared to unconstrained models; in contrast, forcing the model to learn bio-physical concepts might result in a better model overall.

## 4 CONTROLLING AND STEERING

In this Section, we show how the CB-pLM enables fine-grained control, which makes it useful for a variety of protein design tasks. Sequence-based protein design requires the ability to modulate the amount of certain bio-physical properties present in a model's output—a controllable generation problem. For example, we might be interested in controlling a single property, i.e., removing a

hydrophobic patch from a protein, or multiple properties, i.e., increasing the expression and binding affinity of an antibody (Gruver et al., 2024; Tagasovska et al., 2024).

First, we demonstrate how to use CB-pLMs for single and multi-property control. As an example, we instantiate these property control abilities in a Siltuximab—a monoclonal antibody—case study, where we show Grand AVerage of Hydropathy (GRAVY) reduction via CB-pLMs, which alters the protein's surface hydrophobicity.

## 4.1 COMPARING CONDITIONAL LANGUAGE MODEL ARCHITECTURES

Here we assess the effectiveness of the CB-pLM, compared to the alternatives described in Section 3, for both single and multi property concept control.

**Experimental Setup**  We examine 14 biophysical concepts computable from a protein's sequence (e.g., molecular weight, Gravy, Helix fraction, Turn structure fraction; see Appendix B.1). These concepts allow us to measure the success of positive (increase) and negative (decrease) concept interventions for each conditional generation approach. Using a validation dataset, we selected 10,000 sequences with the lowest and highest concept values for positive and negative interventions. To test generation control, we mask 5% of the input sequence (up to 25 amino acids) and intervene on the concept value. For the C-pLM model, masking is random. For CC-pLM and CB-pLM models, we use feature attribution to select tokens to mask (details in Appendix D.1).

Each masked test sample results in a single generated sequence, and change in concept presence, between the original sequence and the generated sequence can be calculated used as estimate of the intervention effectiveness. We use likelihood from an auxiliary autoregressive causal language model to measure the "naturalness of" or feasibility of the sequence (Bachas et al., 2022).

### 4.1.1 SINGLE CONCEPT INTERVENTIONS

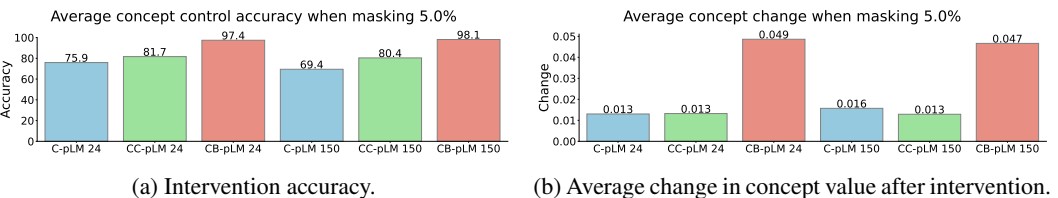

(a) Intervention accuracy.    (b) Average change in concept value after intervention.

Figure 3: Single concept intervention accuracy and effectiveness averaged over all concepts.

**Concept Control Accuracy**  After intervention, we calculate the concept control accuracy as the percentage of generated samples where the change in concept presence is in the direction of the intervention (positive or negative) averaged over a random set of examples (random accuracy would be around 50%). The aggregate average across all concepts is shown in Figure 3a . We find that CB-pLMs outperform other conditional models by over 15%, reaching near perfect accuracy. We also measure the effectiveness of the intervention as the average change in the desired concept value direction; CB-pLM is also $3\times$ higher than other conditional models shown in Figure 3b.

**Change in Concept Distribution**  Beyond comparing average changes, we assess how much the distribution of values shifts upon intervention while maintaining protein naturalness. Ideally, the concept distribution should shift in the direction of the intervention while preserving the naturalness of the protein. Figure 4a illustrates the change in concept and naturalness distributions for a subset of concepts when we intervene on the sequence once. We observe that, for both positive and negative interventions, the CB-pLM effectively shifts the concept distributions in the desired direction for all concepts while preserving the naturalness of the proteins. This performance is superior to other variations of conditional language models. Additionally, we examine the models' ability to iteratively shift a concept's distribution, where the sequence output from the first intervention serves as the input for the second, and so on. Figure 4b shows concept distributions over three iterations for both positive and negative interventions. CB-pLM demonstrates the ability to iteratively shift the concept value with increasing effects, whereas other models fail to achieve this. We refer readers to the Appendix C.1.1 for additional results.

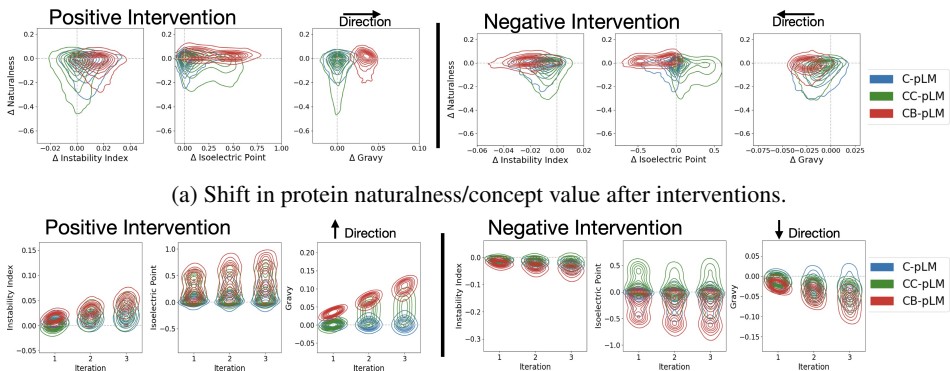

(a) Shift in protein naturalness/concept value after interventions.

(b) Iterative shift in protein concept value after interventions.

Figure 4: Distribution shifts after interventions for different 24M models for subset of concepts.

### 4.1.2 MULTI CONCEPT INTERVENTIONS

Here, we take a step further by demonstrating how CB-pLMs can be used for multi-property optimization—a challenging task in drug discovery (Stanton et al., 2022; Gruver et al., 2024).

**Experimental Setup** Here, we compare the effectiveness of the CB-pLMs to other approaches at enabling control of multiple properties. Specifically, we focus on the concept Grand AVerage of Hydropathy (GRAVY) and the electric charge at pH 7, both standard protein sequence properties correlated with functional properties, such as solubility, viscosity, and aggregation (Bhandari et al., 2020; Cock et al., 2009; Obrezanova et al., 2015). We sequentially intervene on one concept at a time. We test both positive and negative interventions after masking 5% of the sequence as described previously. Similarly, we measure concept control accuracy as the percentage of generated samples that successfully move toward the intervention direction for ***both concepts***. We report the average accuracy for both positive and negative interventions, with random accuracy expected around 25%.

**Result** In Figure 5, we observe that CB-pLM achieves a 94% control accuracy; outperforming the second-best conditional model by 34%. Controlling via the CB-pLM effectively shifts the distributions towards the Pareto frontier, demonstrating its ability to control multiple properties, as shown in Figure 5 (b). We refer readers to Appendix C.1.2 for additional discussion and results.

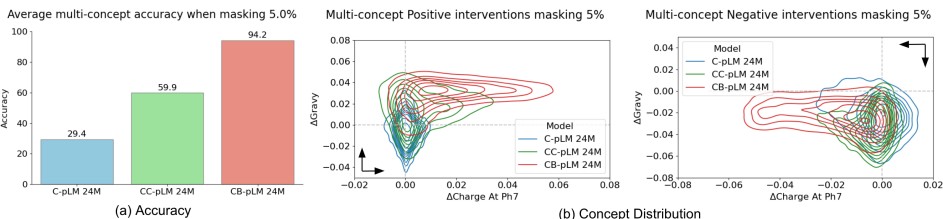

Figure 5: Multi-concept interventions.

### 4.2 CASE STUDY: SILTUXIMAB PROTEIN DESIGN

Siltuximab is an FDA-approved monoclonal antibody used to treat disease in the lymph nodes (Van Rhee et al., 2010). Treatment involves intravenous infusion, which can cause patient discomfort and complications. Re-designing Siltuximab to have more favorable

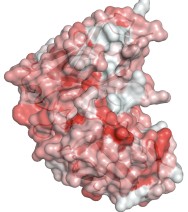

Figure 6: Siltuximab surface hydrophobicity.

biophysical properties is a promising research direction for exploring more effective delivery systems. Structural analysis of Siltuximab reveals the presence of a large hydrophobic patch ( red patch shown in Figure 6), which has been linked to antibody developability liabilities associated

with solubility and aggregation (Bhandari et al., 2020; Obrezanova et al., 2015). Here we test different the effectiveness of the CB-pLM, as well as other conditioning approaches, at redesigning Siltuximab to lower its GRAVY index values, improving its hydrophobicity, so that variants will likely be more soluble.

### 4.2.1 EXPERIMENTS

We train a set of state-of-the-art protein design models on the task of GRAVY reduction, apply the models to Siltuximab, and compare the GRAVY distribution of the generated samples.

**Baselines** We consider a wide variety of representative approaches to guided and unguided sequence generation: (a) **Unguided** Discrete Walk-Jump Sampling (WJS) (Frey et al., 2023) energy- and score-based model with single-step denoising and ESM2 (Lin et al., 2022) a large one-source protein language model. (c) **Implicitly guided** PropEn (Tagasovska et al., 2024) an encoder-decoder architecture that is implicitly trained to optimize a property of interest. (d) **Explicitly guided** LaMBO-2 (Gruver et al., 2024) a classifier-guided discrete diffusion model. (e) **Hydrophilic Resample** a non-deep learning baseline, where residues are randomly resampled from a set of known hydrophilic residues (N, C, Q, G, S, T, Y) (Aftabuddin & Kundu, 2007). Additional details about baselines and training procedures are available in Appendix C.2.

**Results** We intervene on the GRAVY concept for generated samples for each candidate protein design model, with the constraint that generated samples should be at most an edit distance of 5 away from the Siltuximab sequence. In Figure 7a, we show the distribution of designs from each model type. The CB-pLM comes second to LaMBO-2 in terms of the degree of concept distribution shift induced in the generated output, while preserving the naturalness of the protein (Figure 7b). trained to learn over 700 concepts delivers comparable results to state-of-the-art conditioning approaches while preserving the naturalness of the protein.

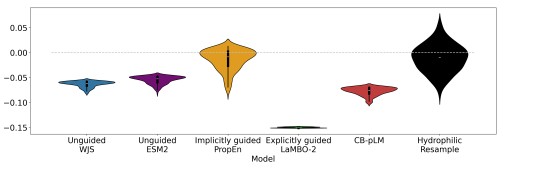
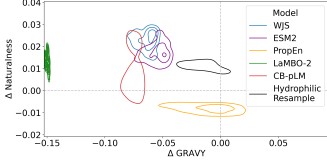

(a) GRAVY distributions.    (b) Naturalness/GRAVY distributions.

Figure 7: Siltuximab redesigns by different models.

## 5 INTERPRETABILITY AND DEBUGGING

Having established the effectiveness of the CB-pLM model for control and steering, demonstrate different *interpretability* and *debugging* capabilities of our model. A key component of the CB-pLM is its final linear layer—a deliberate choice to maintain simplicity. By inspecting the weights of this linear layer, domain experts can understand the impact of specific concepts on the model's output. Moreover, if the model learns undesirable correlations or if the learning process results in important features that deviate from a domain expert's prior knowledge, a straightforward inspection of the linear layer's weights can help identify and address these issues.

### 5.1 INTERPRETABILITY

CB-pLM offers two types of interpretability: Local and Global. Local Interpretability allows the model to indicate which concepts it relies on for predictions and identify the input tokens that influence its output, achieved through our concept-bottleneck module and feature masking and regulation scheme. Global Interpretability provides insights into the relationships between predefined concepts and various tokens, facilitated by the linear decoder. This section explores CB-pLM's global interpretability and verifies its alignment with human knowledge.

In Figure 8, we show the final layer weights for different concepts and amino acids in our CB-pLM (24M). We find that the model successfully learns several key biophysical relationships. We highlight some key insights below (the full weight matrix and more details are available in Appendix D):

- For both charge at pH 6 and pH 7, acidic amino acids (D, E) have negative weights, while basic amino acids (R, K) have positive weights. The difference in charge between pH 6 and pH 7 reflects the biophysical properties of Histidine (H), as it contains an imidazole side chain with a pKa of 6.0 and contributes more towards positive charge at lower pH (Figures 8a, 8b).
- GRAVY, which defines hydropathy based on the Kyte-Doolittle scale (Kyte & Doolittle, 1982), is consistent with positive weights assigned to A, I, V, F, C, and M amino acids (Figure 8c ).
- Aromatic amino acids F, Y, and W have high weights for the aromaticity concept (Figure 8d).

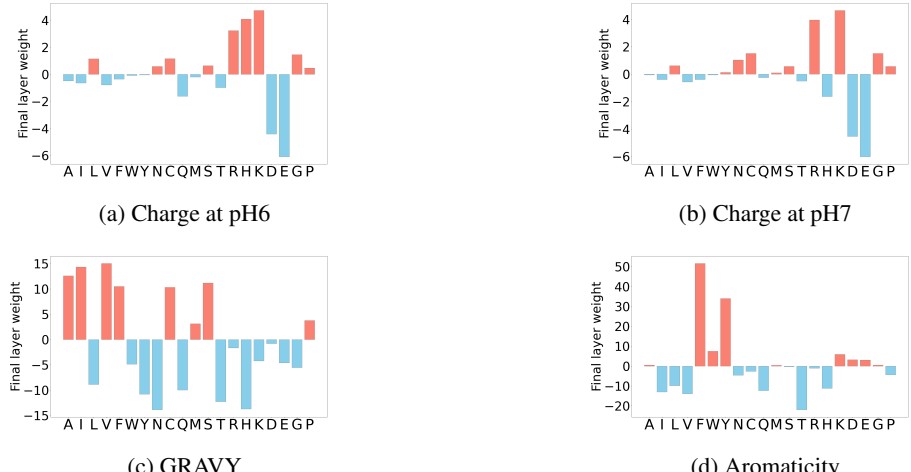

(a) Charge at pH6

(b) Charge at pH7

(c) GRAVY

(d) Aromaticity

Figure 8: The weights for selected concepts to amino acids in the last layer CB-pLM 24M.

## 5.2 DEBUGGING

Model debugging involves detecting and fixing errors during training or testing. Identifying the root cause of an error–such as issues with optimization, initialization, training data, or function class–can be challenging and often relies on manual effort or retraining. CB-LM introduces mechanisms to attribute errors to human-understandable concepts, enabling us to answer key questions: 1) Did the model learn an important feature of interest? 2) Is the model relying on a spurious feature? 3) In which situations is the model likely to fail? 4) How can we correct unwanted correlations? These questions can be addressed by inspecting the validation loss of different concepts, the weights of the linear layer, and using the CB layer for interventions.

To showcase the model debugging capability of a CB-pLM, we trained a *Bad* CB-pLM 24M, where we change the normalization for the aromaticity, making it very difficult for the model to learn and control it. We want to see if we can, in fact, identify where this model will fail without the need for intervention experiments. To do this, we examine the weight for aromaticity-amino acid combinations for the Bad CB-pLM (see Figure 9a). By inspection, we observe that weights are all essentially zero; hence, the model will not be able to control aromaticity. To confirm the result, in Figure 9b, we show that intervening on the aromaticity concept does not control the model's output.

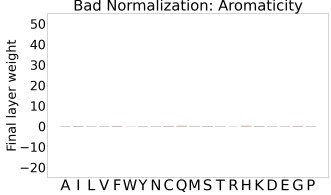

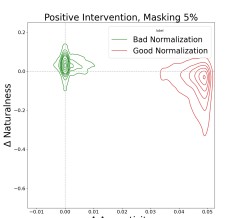

(a) Outgoing weights from aromaticity for model with bad concept normalization.

(b) The effect of intervention on the aromaticity with good vs bad model.

Figure 9: CB-pLM linear layer helps identify model's failures.

## 6  DISCUSSION AND CONCLUSION

**Related Work** Language models are increasingly used across various fields, demonstrating exceptional performance but lacking interpretability, which limits their use in high-stakes scenarios. While recent work with sparse autoencoders (SAEs) Cunningham et al. (2023); Templeton et al. (2024) shows promise in uncovering interpretable features, this approach requires training large SAEs and interpreting millions of features. Concept bottleneck language models (CB-LMs) offer a simpler solution by directly encoding concepts without additional interpretation steps.

Concept Bottleneck Models enhance neural network interpretability by incorporating a "concept bottleneck" layer that maps inputs to human-understandable concepts for final predictions (Koh et al., 2020). Recently, Ismail et al. (2023) extended this approach to generative models (CBGMs), enabling controllable generation and concept-level explanations, particularly in image generation tasks. In this paper, we adapt this approach to generative language models, achieving global control over the entire input while retaining token-level generation. A few very recent works have proposed CBMs for text classification tasks Tan et al. (2024); Sun et al. (2024), but this is a much easier and less interesting setting than generative language modeling which we address.

Protein language models (pLMs) are extensively used in biological machine-learning research. Despite their critical applications in healthcare and drug discovery (Hie et al., 2024), pLMs currently lack interpretability. Historically, pLM architectures, loss functions, and training setups have closely mirrored the original masked language model setup (Devlin et al., 2018), differing mainly in vocabulary and dataset. While some protein language models support conditional generation by concatenating different protein functions and properties to the inputs (Shuai et al., 2021; Madani et al., 2020; Hayes et al., 2024), they do not provide mechanisms for interpretability, debuggability, or insights into what the model has learned. See Appendix E.1 for extended related work.

**Discussion and Future Work** In this paper, we applied our proposed architecture to masked language models for proteins. This architecture can be adapted to any domain, such as NLP, by changing the training dataset, vocabulary, and concepts while keeping the model unchanged. It can also be extended to other language models, such as autoregressive models (Figure 29 in Appendix E.2). In principle, CB-LM can be used to generate sequences with unseen concepts or novel combinations, as long as they can be expressed as functions of existing ones. We demonstrate this capability in a toy task detailed in Appendix C.3.2. Testing this capability empirically on proteins is beyond the scope of the current paper and is left for future work. Our method is unlikely to enable the development of weaponizable molecules due to limited biophysical understanding of biological functions and physiochemical properties. However, we stress the importance of biosecurity and recommend consulting regulatory agencies and ethical guidelines to mitigate potential risks.

**Limitations** One main limitation of CB-LM, like other concept bottleneck models (Ismail et al., 2023; Koh et al., 2020), is the need for the training dataset to be annotated with predefined concepts. Another drawback is the necessity for unknown embeddings. Generally, we believe this is unavoidable, as it is impractical to encode all the information needed for a generative task into a fixed number of predefined concepts; excluding unknown embedding will lead to unacceptable performance losses. However, reliance on unknown parts can challenge control and interpretability, potentially causing the model to ignore concept outputs. Our method mitigates this with orthogonality loss to improve control, but some concept information can remain in the unknown part, and better disentanglement methods are needed.

**Conclusion** We present Concept Bottleneck Language Models (CB-LM), a ***controllable-interpretable-debuggable*** language model. We have demonstrated the effectiveness of our proposed architecture by applying it to masked protein language models. Our results show that this architecture can scale to large models, as evidenced by training Concept Bottleneck Protein Language Models (CB-pLM) with 24M, 150M, 650M, and 3B parameters to encode over 700 human-understandable concepts while maintaining pre-training perplexity comparable to traditional masked protein language models. Our findings indicate that CB-pLM offers superior control compared to conditional language models, along with interpretability and debugging capabilities. This work represents a significant step towards more interpretable, reliable, and debuggable neural networks.

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

# Appendix

## A  ADDITIONAL DETAILS ON CONCEPT BOTTLENECK LANGUAGE MODEL

### A.1  LOSS FUNCTION

Following CBGMs (Ismail et al., 2023), concept bottleneck language models are trained with three losses: a standard generative masked language modeling loss $\mathcal{L}_{\text{MLM}}$, concept loss $\mathcal{L}_{\text{Concept}}$ and an orthogonality loss $\mathcal{L}_{\text{orth}}$. The final loss is given by

$$\mathcal{L}_{\text{total}} = \mathcal{L}_{\text{MLM}} + \alpha \mathcal{L}_{\text{Concept}} + \beta \mathcal{L}_{\text{orth}},$$

where $\alpha$ and $\beta$ are hyperparameters.

- *Generative masked language modeling loss:*

$$\mathcal{L}_{\text{MLM}} = -\mathbb{E}_{x,m} \left[ \frac{1}{m} \sum_{i \in m} \log P(x_i \mid x_{\backslash m}) \right],$$

  where $m$ is the set of positions of the randomly masked tokens, and the model is trained to predict the identity of the masked tokens.

- *Concept loss:* Given the general-purpose nature of large language models, they are expected to handle thousands of concepts with categorical or real values. Often, samples lack values for most concepts. To address this, we normalize all real-valued concepts to [0, 1] and apply mean square error loss on the entire concept embedding:

$$\mathcal{L}_{\text{Concept}} = \frac{1}{k} \sum_{i=1}^{k} (c_i - \hat{c}_i)^2.$$

  Missing values are replaced with default values, and their effect is removed from the loss function by considering only non-missing concepts, achieved by masking the errors before backpropagation.

- *Orthogonality loss:* Following Ismail et al. (2023), to ensure effective control over the model's output, it is crucial to prevent unknown concepts from being mere transformations of known concepts. This is achieved by enforcing an orthogonality constraint (Ranasinghe et al., 2021), which minimizes the cosine similarity between the concept context embedding and the unknown context embedding. The orthogonality loss is defined as follows:

$$\mathcal{L}_{\text{orth}} = \sum_{j \in B} \frac{\sum_{i=0}^{i=s} \left| \langle z, \widetilde{h}_i \rangle \right|}{\sum_{i=0}^{i=s} 1} \tag{1}$$

  where $\langle \cdot, \cdot \rangle$ is the cosine similarity applied to two embedding, $| \cdot |$ is the absolute value, and $B$ denotes mini-batch size and $j$ denotes each sample in the mini-batch. The cosine similarity in the above equation involves the normalization of features such that $\langle x_i, x_j \rangle = \frac{x_i \cdot x_j}{\|x_i\|_2 \cdot \|x_j\|_2}$, where $\| \cdot \|_2$ is $l_2$ norm.

### A.2  TRAINING FOR INTEPRETABILITY

**Reducing concept leakage through independent training:** One known problem in CBMs is concept leakage (Margeloiu et al., 2021; Mahinpei et al., 2021; Havasi et al., 2022); this happens when soft concept representations encode more information than the concepts themselves. Concept leakage affects both model interpretability and control. Mahinpei et al. (2021) discussed that this leakage happens when CBMs are trained jointly or sequentially. However, leakage is not possible when CBMs are trained independently (i.e., during training, the post-CB-layer part of the network takes the ground-truth concepts themselves rather than the output of the CB-layer). For this reason, we choose to use independent training for CB-LM.

**Faithful token attribution through regularization:** Apart from concept-level explanations, users might be interested in token-level explanations to identify which tokens were most influential during generation. Token-level explanations can also be used to identify which token to mask (i.e., the token that has the most effect on a particular concept) when intervening on different concepts (Gruver et al., 2024). Gradient-based explanation methods Baehrens et al. (2010); Simonyan et al. (2013b); Zeiler & Fergus (2014) are popular approaches for 'explaining' models, but many works (Adebayo et al., 2018; Hooker et al., 2018) showed that such feature attribution methods are no better than random ranking of the input features. The reliability of feature attributions seems to be correlated to how the model is trained; Shah et al. (2021) showed that gradient-based feature attributions of adversarially robust image classifiers are faithful, unlike their non-robust counterparts; Adebayo et al. (2023), showed that the addition of Gaussian noise to the input has the same effect as adversarial training for inducing attribution faithfulness. Following Adebayo et al. (2023)'s recommendation, we add Gaussian noise to the token embedding during training for model regularization and faithful token attribution.

# B    ADDITIONAL DETAILS ON CONCEPT BOTTLENECK PROTEIN LANGUAGE MODEL

## B.1    TRAINING DATA

We combined sequences from UniRef50 (Suzek et al., 2015) and SWISS-PROT (Bairoch & Apweiler, 2000), removing duplicates. Annotations such as protein clusters, organisms, taxons, biological processes, cellular components, and molecular functions from SWISS-PROT were used as concept annotations. Biopython (Cock et al., 2009) was used to extract biophysical and bioinformatics sequence-level concepts. We filtered out rare SWISS-PROT concepts; the final list of concepts is shown in 2.

| Class | Number of Concepts | Calculated from sequence |
|---|---|---|
| Cluster name | 159 | ✗ |
| Biological process | 140 | ✗ |
| Cellular component | 162 | ✗ |
| Molecular function | 106 | ✗ |
| Organism | 36 | ✗ |
| Taxon | 101 | ✗ |
| Biopython | 14 | ✓ |

Table 2: Training concepts.

**Biopython**    All concepts were computed from the Biopython ProtParam module.

- **Molecular weight** is the mass of the protein.
- **Charge at pH 6 and pH 7** is the net charge of the protein at pH 6 and pH 7.
- **Isoelectric point** is the pH at which the protein has no net charge.
- **Aromaticity** is the relative frequency of aromatic amino acids (F, W, Y) in the sequence
- **Instability Index** is the sum of the instability weights from sequence decomposition.
- **Secondary structure fraction** is the relative frequency of amino acids in helix: V, I, Y, F, W, L. Amino acids in Turn: N, P, G, S. Amino acids in sheet: E, M, A, L.
- **Molar extinction coefficient** considers the weighted sum of W, Y, and C for protein absorption of 280 nm.
- **Gravy** is the sum of hydropathy values associated with the amino acids in a sequence divided by the total number of amino acids, based on the Kyte and Doolittle hydrophobicity scale.
- **Protein Scale** An amino acid scale is defined by a numerical value assigned to each type of amino acid. The most frequently used scales are the hydrophobicity or hydrophilicity scales, the secondary structure conformational parameters scales and surface accessibility.

## B.2 MODEL PARAMETERS AND CONFIGURATIONS AT DIFFERENT SCALE

|                        | 24M    | 150M   | 650M   | 3B                |
|------------------------|--------|--------|--------|-------------------|
| Number of layers       | 10     | 27     | 33     | 26                |
| Embedding dim          | 408    | 768    | 1280   | 2560              |
| Attention heads        | 12     | 12     | 20     | 40                |
| Concept embedding dim  | 2      | 2      | 2      | 2                 |
| Learning rate          | 0.001  | 0.0001 | 0.0001 | 0.0001            |
| Clip norm              | 0.5    | 0.5    | 0.5    | 0.5               |
| precision              | 16     | 16     | 16     | bf16              |
| Warmup steps           | 3000   | 10000  | 30000  | 30000             |
| Effective batch size   | 512    | 1024   | 1024   | 1024              |
| Distributed backend    | ddp    | ddp    | ddp    | deepspeed_stage_1 |

**Training:** During training we mask percentage is 25% of the sequence. We then truncate all sequences to a maximum length of 512. Sequences of length less than 512 were padded, but no loss was backpropagated through the network for padding tokens. We use Rotary Position Embedding.

## C CONTROL EXPERIMENTS

### C.1 COMPARING CONDITIONAL LANGUAGE MODEL ARCHITECTURES *in silico*

**Experimental Setup** We evaluate control on 14 different concepts that can be calculated from Biopython. We want to see if we can intervene in the models by increasing the concept values (positive interventions) or decreasing the concept values (negative interventions). To accomplish this, we utilized the held-out validation dataset; for each concept, we selected the 10,000 sequences with the lowest concept values and the 10,000 sequences with the highest concept values. We use the lowest value sequences for positive interventions (i.e., we intervene in the model to increase the value of that given concept), and the highest value sequences are used for negative interventions (i.e., we intervene to decrease the concept value). We mask a percentage of the test sequence and intervene on the concept to set it to a minimum or maximum value based on the intervention type. For the conditional model, C-pLM, masking is done randomly; for the conditional classifier CC-pLM and our proposed CB-pLM model we select the tokens that contributed to the given concept the most using feature attribution (Gruver et al., 2024; Adebayo et al., 2023) (additional details on how feature attribution is done is available in Appendix D.1) and mask them. For each test sample, we only consider a single generated sample (i.e this can be viewed as single shot generation). After generation, we calculate the new concept values of the generated sequence. We measure accuracy as the percentage of generated samples that successfully moved toward the direction of the intervention. We use likelihoods from an auxiliary autoregressive causal language model to measure the "naturalness of" or feasibility of the sequence (Bachas et al., 2022).

#### C.1.1 SINGLE CONCEPT INTERVENTION

In this section, we focus on single-concept intervention; the intervention procedure is as follows:

ONE TIME INTERVENTION

This sections shows results when setting $n = 1$ in Algorithm 1.

**Accuracy** After the intervention, we calculate the accuracy as the percentage of generated samples that successfully moved toward the direction of the intervention for positive and negative interventions, then average them (random accuracy would be around 50%). The average across per concept is shown in Figure 10b. Our findings demonstrate that in terms of intervention direction accuracy, CB-pLM outperforms other conditional models on *every single concept*. Figure 11b shows the average change in the desired direction of the concept value of different concepts after our interventions. This is averaged over positive and negative interventions. The concept values are normalized such that the maximum value in training data is 1 and the minimum value is 0. We can see our CB-pLM models clearly outperform baselines for almost all concepts.

---

**Algorithm 1** Single Concept intervention procedure

---

**Require:** sequence to intervene on $x = [x_0, ..., x_s]$, where $s$ is the maximum sequence length, model (i.e, CB-pLM) $f$, concept to intervene on $k$, concept value after intervention $c$, direction of intervention $d$ and number of interventions $n$.

1: $i \leftarrow 1$
2: **while** $i \leq n$ **do**
3:     Using feature attribution, identify the top 5% of the token that either negatively influences the concept for positive intervention or positively influences the concept of negative intervention. $x_{\text{masked}} \leftarrow masking(x, k, f, d)$
4:     Pass the model the masked input, the concept index, and the value to get the new sequence. $\bar{x} = f(x_{\text{masked}}, k, c)$
5:     $x \leftarrow \bar{x}$
6: **end while**
7: **return** $x$

---

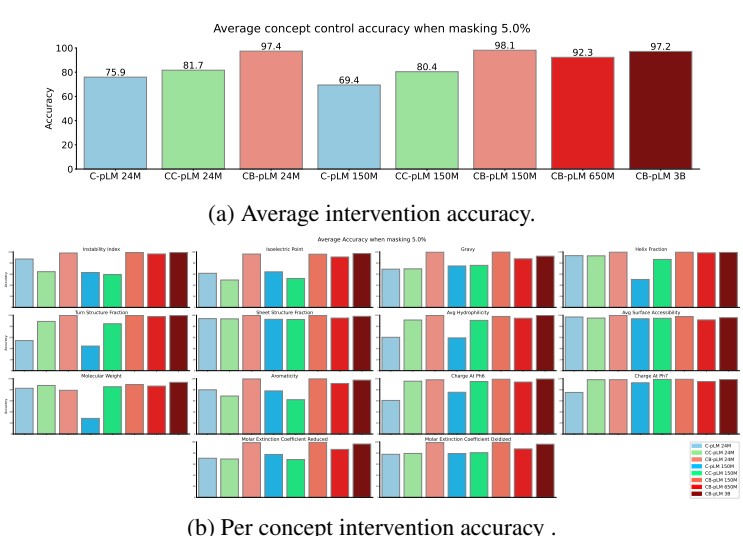

(a) Average intervention accuracy.

(b) Per concept intervention accuracy .

Figure 10: Intervention accuracy after masking 5%.

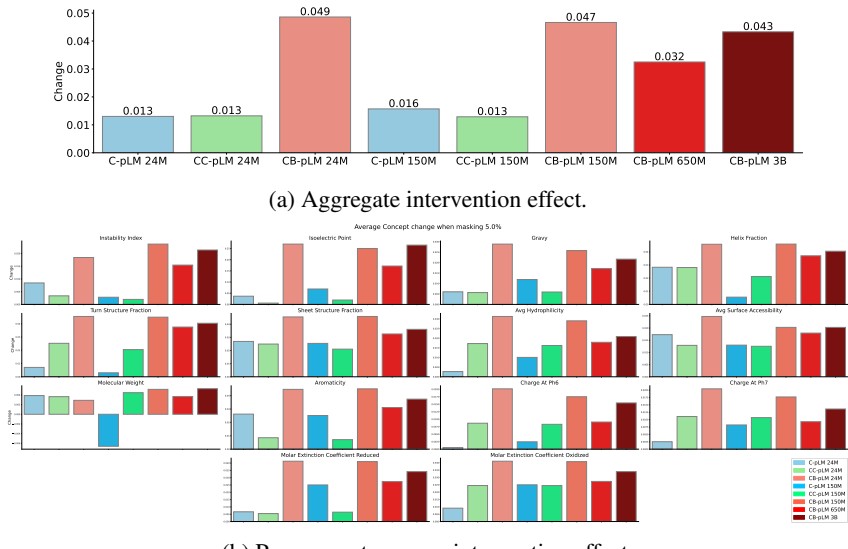

(a) Aggregate intervention effect.

(b) Per concept average intervention effect.

Figure 11: Intervention effect after masking 5%.

**Distribution shift** We investigate how the concept distribution shifts along with the naturalness of the protein after intervention. Ideally, we would want the concept distribution to shift in the direction of the intervention while maintaining the naturalness of the protein; we plot $\Delta$ Concept/Naturalness distribution (the difference between the original sample and the generated sample for the concept value on $x$-axis and naturalness on $y$-axis).

Figure 12 and Figure 13 shows distribution shifts between different types of 24M and 150M models respectively, with both positive and negative interventions, CB-pLM effectively shifts the concept distributions in the right direction for all concepts while preserving the naturalness of the proteins, unlike other variations of conditional language models. Figure 14 shows distribution shifts between CB-pLM with different sizes. Note that although increasing model size improved the model's perplexity we did not observe any improvement in terms of control.

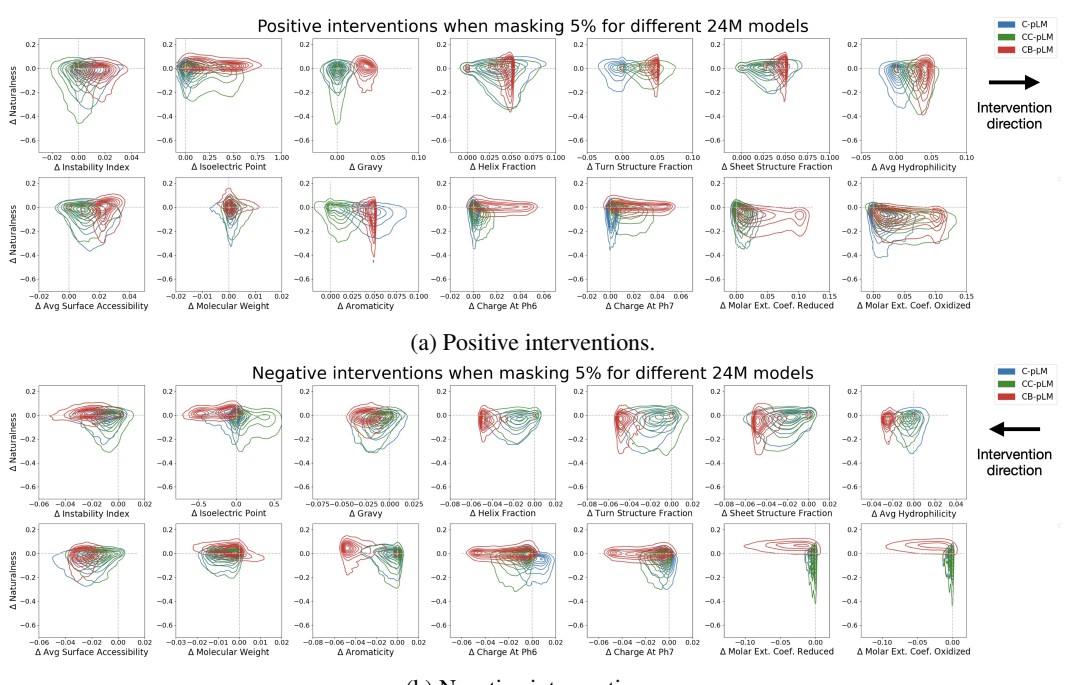

(a) Positive interventions.

(b) Negative interventions.

Figure 12: Shift in protein naturalness/concept value after interventions for different 24M models.

**Iterative intervention** In this experiment, we examine different models' ability to shift a concept's distribution iteratively, i.e.; we vary $n$ in Algorithm 1 from $1 \rightarrow 3$. Figures 15 and 16 shows concept distributions for three iterations for both positive and negative interventions. CB-pLM can iteratively shift the concept value with increasing effects while other models fail to do this.

**Masking percentage** In this experiment, we investigate the impact of varying the percentage of masking on the performance ranking of different models. We repeat the experiment described in Algorithm 1, but with 25% masking instead of 5%, using 24M parameter models. Our findings show that the performance ranking of the models remains consistent, with CB-pLM significantly outperforming both C-pLM and CC-pLM (Figures 17,18 and 19).

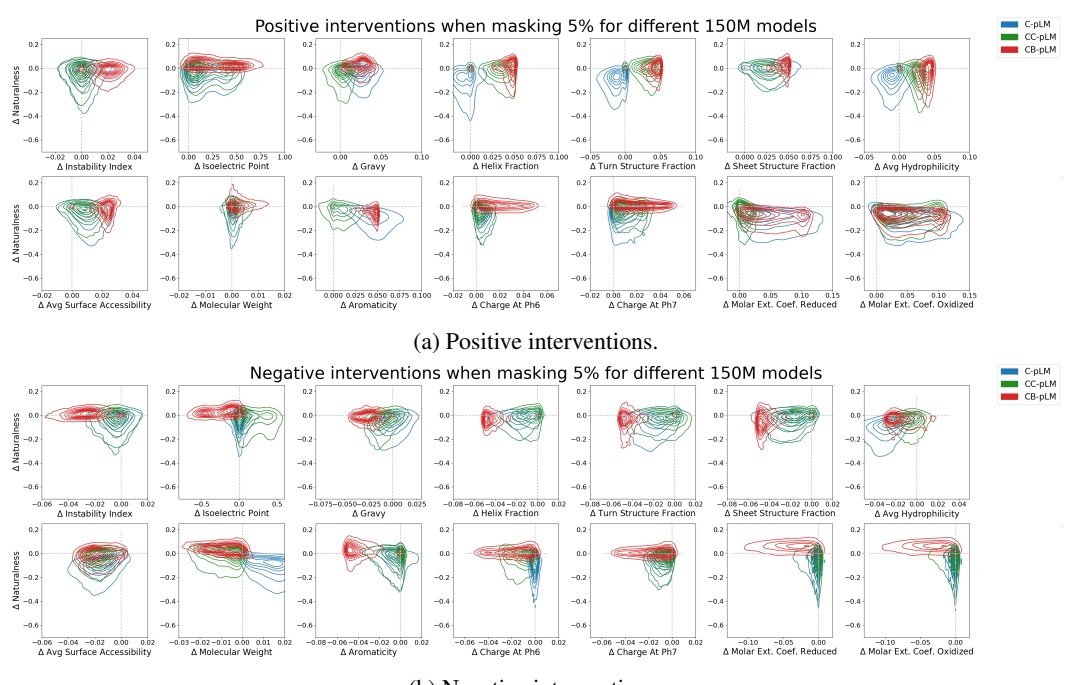

(a) Positive interventions.

(b) Negative interventions.

Figure 13: Shift in protein naturalness/concept value after interventions for different 150M models.

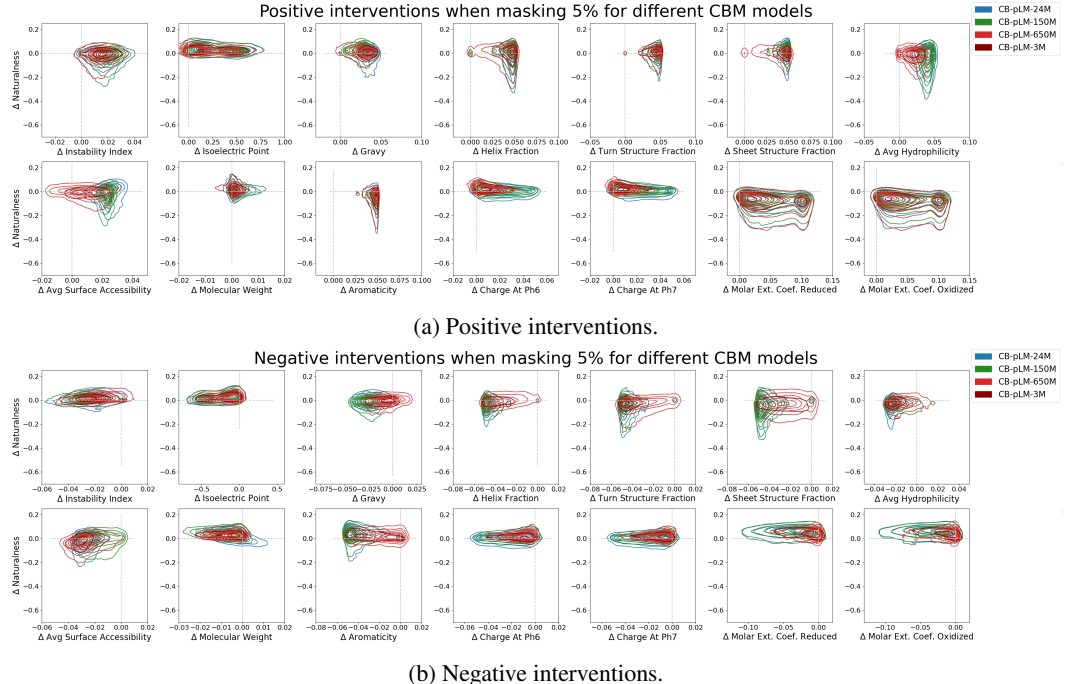

(a) Positive interventions.

(b) Negative interventions.

Figure 14: Shift in protein naturalness/concept value after interventions for CB-pLM models with different sizes.

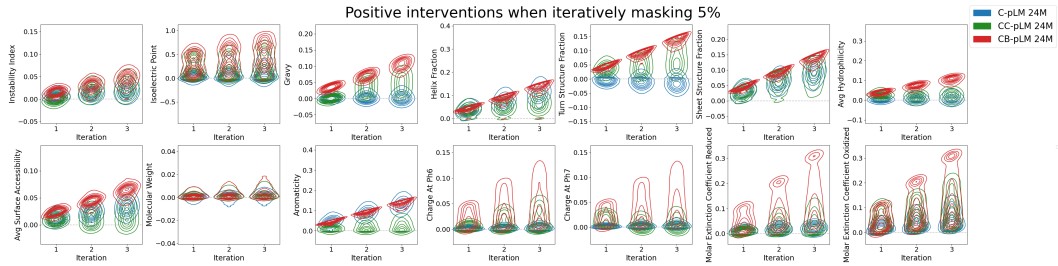

Figure 15: Iterative positive intervention.

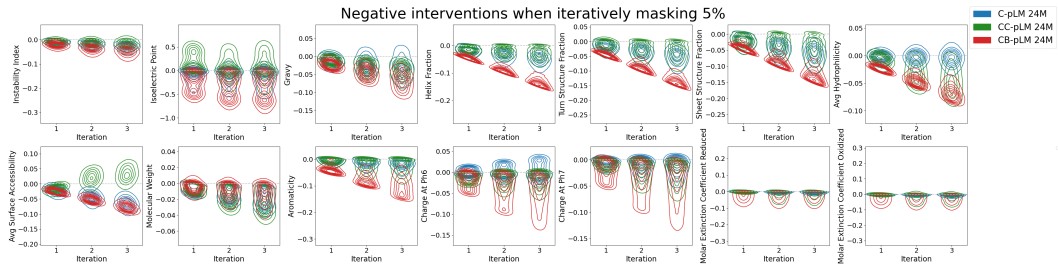

Figure 16: Iterative negative intervention.

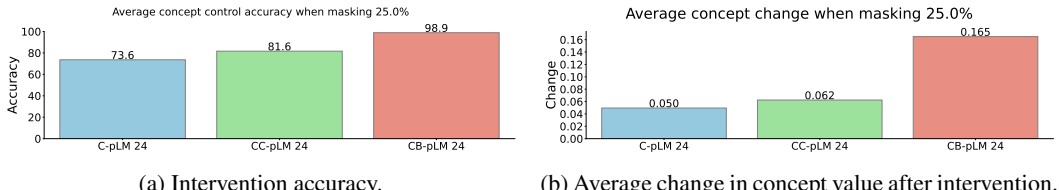

(a) Intervention accuracy.

(b) Average change in concept value after intervention.

Figure 17: Average concept intervention accuracy and effectiveness when masking 25%.

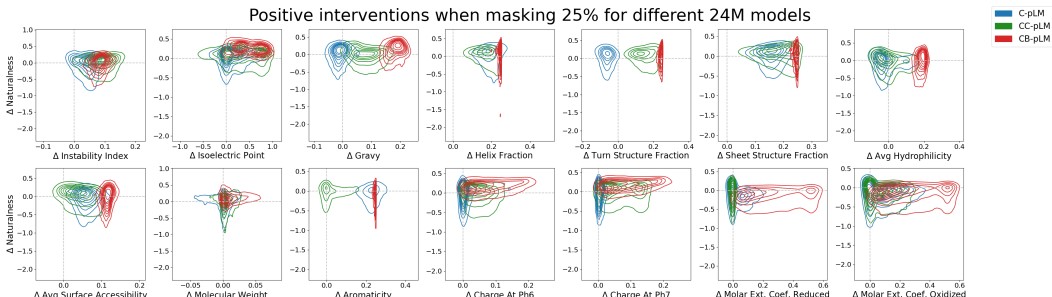

Figure 18: Shift in protein naturalness/concept value for positive interventions when masking 25%.

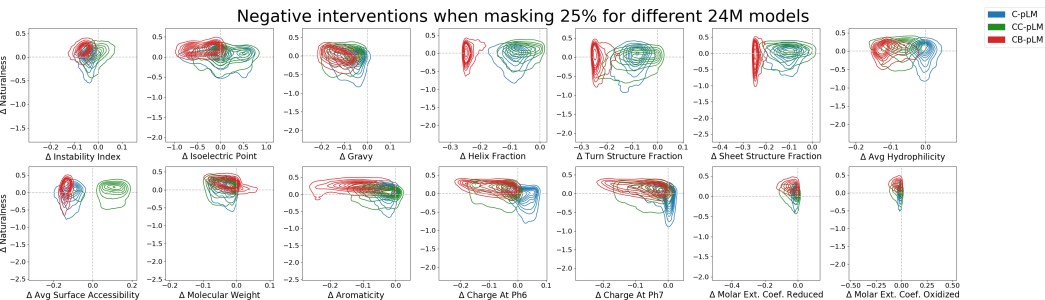

Figure 19: Shift in protein naturalness/concept value for negative interventions when masking 25%.

**Concept intervention correlation**  In this experiment, we mask the most influential tokens for each concept as previously described and intervene on the value of that concept. We then calculate the value of all 14 concepts for the generated sequence, and see how intervening on one concept effects all the other concepts. The results are reported in Figures 20b, 20c and 20d for different models. In this figure we report a version of accuracy, where the $j$-th element of the $i$-th row represents the (fraction of times concept $j$ changed in the same direction as the intervention when intervening on the $i$-th concept) - (fraction of times concept $j$ changed in the opposite direction of the intervention when intervening on the $i$-th concept). This can be seen as a binary version of correlation. We also compare this with the actual correlation between these concept values on existing proteins in our dataset shown in Figure 20a. We can see the correlation matrix of interventions on our CBM has many similarities to the correlations between the concept in the real world, especially on pairs with high correlation, reflecting that our concept bottleneck model has picked up on these natural correlations. In contrast, the other models have a quite different correlation pattern.

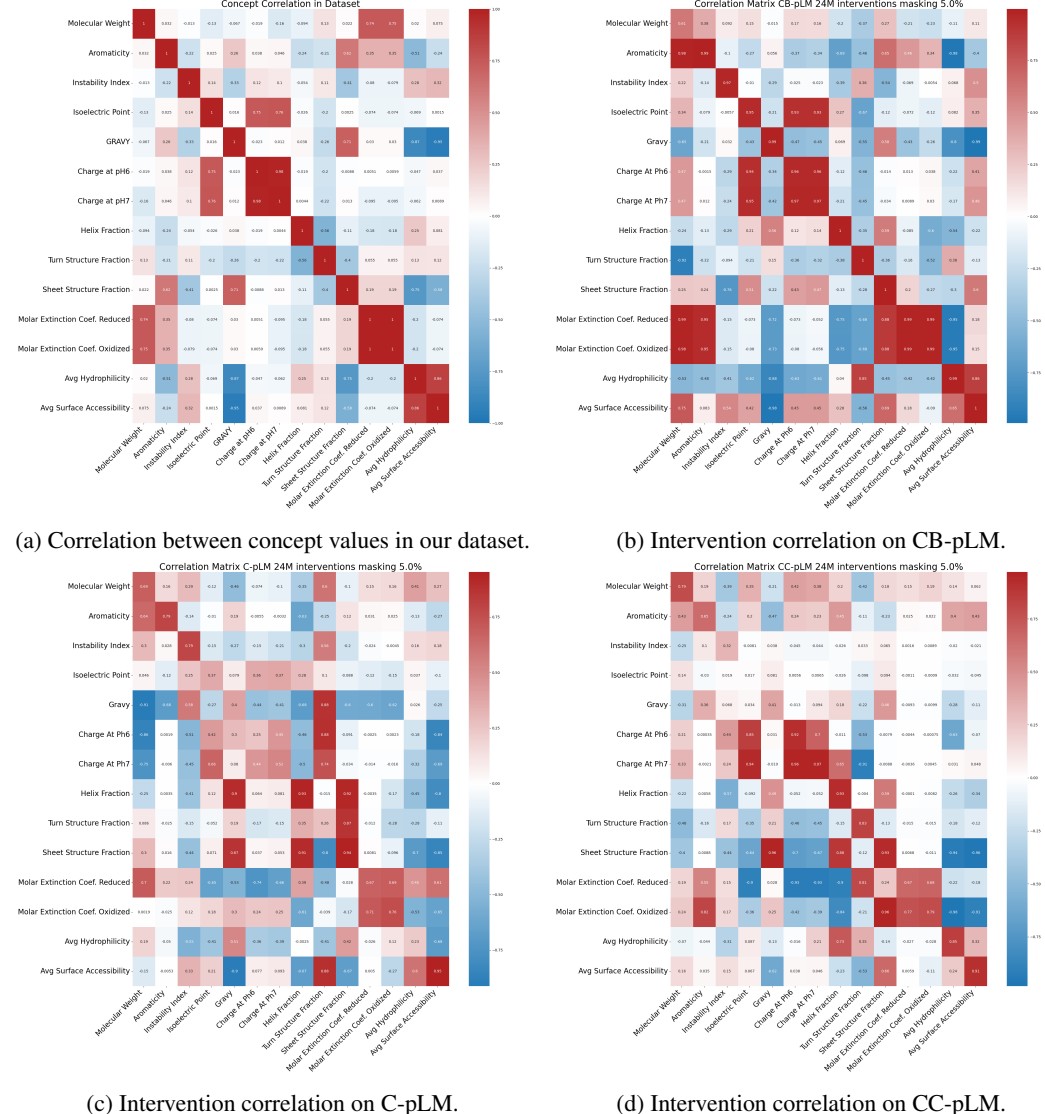

(a) Correlation between concept values in our dataset.

(b) Intervention correlation on CB-pLM.

(c) Intervention correlation on C-pLM.

(d) Intervention correlation on CC-pLM.

### C.1.2 MULTI-CONCEPT INTERVENTION

We show the effects of sequentially intervening on two concepts in Figure 21 with different 150M parameter models below.

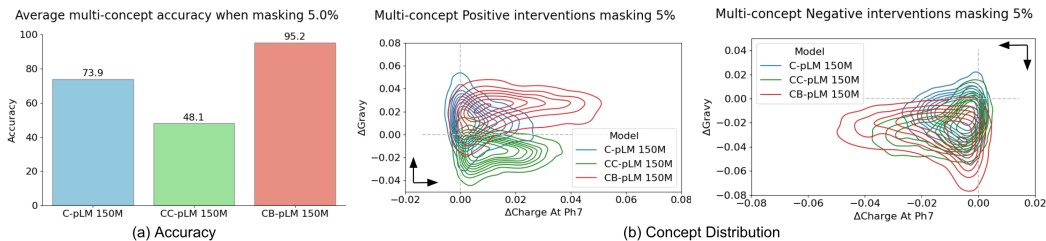

Figure 21: Multi-concept interventions.

## C.2 PROTEIN DESIGN - SILTUXIMAB

### C.2.1 BASELINES

- **LaMBO-2** (Gruver et al., 2024) is diffusion optimized sampling, a guidance method for discrete diffusion models that follows gradients in the hidden states of the denoising network. LaMBO-2 was trained on SWISS-PROT ($\sim$ 2.6M training samples training) using GRAVY as the optimization objective for explicit guidance.
- **PropEn** (Tagasovska et al., 2024) is an encoder-decoder architecture that is implicitly trained to optimize a property of interest. PropEn operates by matching training samples so that each training sample is paired with one that has better properties; due to this matching, it is best suited for a low data regime. We train PropEn on a curated antibody dataset of $\sim$ 1000 training samples that are close to Siltuximab.
- **Discrete Walk-Jump Sampling (WJS)** (Frey et al., 2023) is a discrete generative model that learns a smoothed energy function, then samples from the smoothed data manifold with Langevin Markov chain Monte Carlo (MCMC), and projects back to the true data manifold with one-step denoising. WJS was trained on paired observed antibody space (OAS) dataset Olsen et al. (2022), which is $\sim$ 120K paired antibody sequences as done in the original paper.
- **ESM2** (Lin et al., 2022) an open source 150M parameter protein language model, pretrained on UniRef50. Designs are sampled from the model's logits.
- **Hydrophilic Resample** a non-deep learning baseline, where residues are randomly resampled from a set of known hydrophilic residues (N, C, Q, G, S, T, Y) (Aftabuddin & Kundu, 2007).

### C.2.2 ASSESSING THE NATURALNESS OF THE DESIGNS

To evaluate the naturalness of designs generated by various models, we folded the designs using ABodyBuilder2 (Abanades et al., 2023) and analyzed different protein surface properties with the Therapeutic Antibody Profiler (TAP) Raybould & Deane (2022), which utilizes physics-based computations. Our objective is to achieve minimal deviations from the original antibody structure. Figure 22 presents the TAP scores of different designs, with the dotted line indicating the original Siltuximab value. The results reveal that the designs produced by CB-pLM and WJS exhibit the least disruption, whereas those from ESM2 and Hydrophilic Resample show the most disruption. The TAP metrics validate that our CB-pLM designs effectively preserve naturalness.

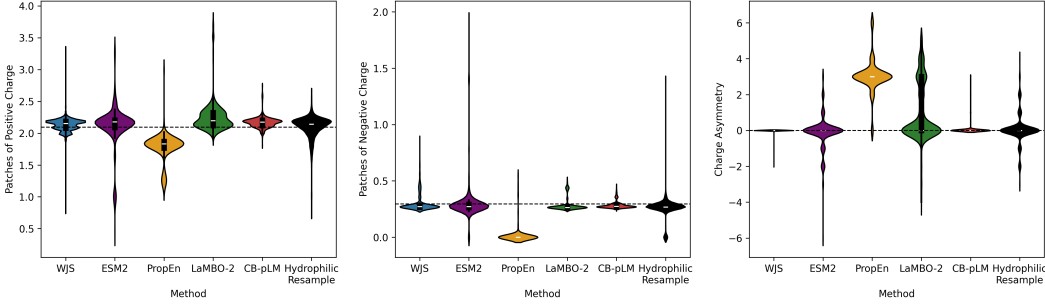

Figure 22: TAP metrics for Siltuximab redesigns.

**Filtering Designs** Out of the designs produced by different models, we have filtered designs with TAP values that are with 5% proximity to Siltuximab; the percentage of designs that have passed this filter from each model is shown in Figure 23.

**Figure 23:** Percentages of redesigns within the naturalness threshold.

## C.3 ADDITIONAL CONTROL EXPERIMENTS

### C.3.1 SPECIES INTERPOLATION

Prior work in mechanistic interpretability (Bricken et al., 2023) demonstrated that sparse autoencoders (SAEs) could uncover interpretable monosemantic features. Recently, Templeton et al. (2024) applied SAEs to Claude 3 Sonnet's activations, retrieving $\sim 34M$ features. They used the autointerpretability approach (Bills et al., 2023) to assign meanings to these features. One notable feature identified was the *Golden Gate Bridge*. By clamping this feature to $10\times$ its maximum activation, they influenced the model's behavior, causing Claude to behave in an out-of-distribution manner, going as far as self-identifying as the Golden Gate Bridge. Here, we show how the concept bottleneck language model demonstrates the same capability, although for CB-LMs *there is no need to train an additional SAE and do a post-hoc search for the meaning of different concepts, since here we know what each concept encodes by design*.

To demonstrate, we test the CB-pLM's ability to interpolate between species; we altered the behavior of the CB-pLM by setting a single species concept in the original CB-pLM to 10 times its maximum activation value. We then tested whether the model could generate a sequence similar to that species cluster, regardless of the protein sequences inputted into the model. We conducted this test for two species: (a) rice (species *Oryza sativa* - referred to as "Rice-CB-pLM") and (b) human (species *Homo sapiens* - referred to as "Human-CB-pLM"). We trained a species classifier by finetuning the ESM2(Lin et al., 2022) model to classify different species (humans, yeast, rice, and bacteria). The model accurately clustered different species together, with an overall accuracy of 98%, as shown in Figure 24a. We then passed non-rice protein sequences into Rice-CB-pLM and classified the output; most newly generated sequences were now identified as rice (Figure 24b). Similarly, passing non-human sequences into Human-CB-pLM resulted in most newly generated sequences being classified as human (Figure 24c); note that changing non-human protein sequences into human sequences is a fundamental problem in protein engineering known as "humanization".

This illustrative example showcases the ability of the concept bottleneck language model to replicate the capabilities of SAEs and autointerpretability with a substantially simpler approach.

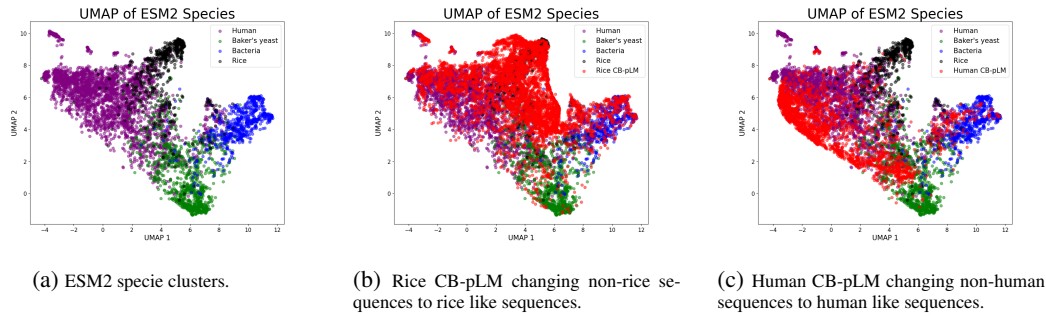

(a) ESM2 specie clusters.

(b) Rice CB-pLM changing non-rice sequences to rice like sequences.

(c) Human CB-pLM changing non-human sequences to human like sequences.

Figure 24: Manipulating Large Language Models.

### C.3.2 GENERALIZATION TO NEW CONCEPT COMBINATION

So far we have shown that our model can successfully control generation of in distribution sequences. However, for many real-world use cases, it is important to be able to generate previously unseen concepts or combinations of concepts.

To test this, we train a model with our architecture on a simpler task based on color-MNIST. To make this task similar to masked language modeling, we train our models by masking 75% of the input, and then train them to reconstruct the remaining input using a masked autoenecoder vision transformer (MAE-ViT) (He et al., 2022). Following MAE-ViT, we divide the input into a sequence of 7x7 patches, making our task a masked sequence-to-sequence generative task, just like masked language modeling. Note that unlike our main results, we use a nonlinear decoder to accommodate for pixel-level outputs. For the concepts in this experiment, we use a one-hot encoding of the number represented by the image, together with an RGB encoding of the color of the input digit. It is important to use RGB embeddings instead of one-hot representation of color, as it allows to encode any color in just 3 dimensions, which is useful for testing generalization capabilities. To test the generalization ability of our CBM architecture, we trained a generative CBM as discussed above on ColorMNIST with 7 discrete training colors: {magenta, grey, cyan, white, red, green, yellow}. On test time, we then intervened on the CBM color outputs to make the model create an unseen color. As shown in Figure 25, our method can accurately regenerate the the input for all 3 unseen colors we tested. These examples were not cherry picked, and our color intervention reliably works on all inputs.

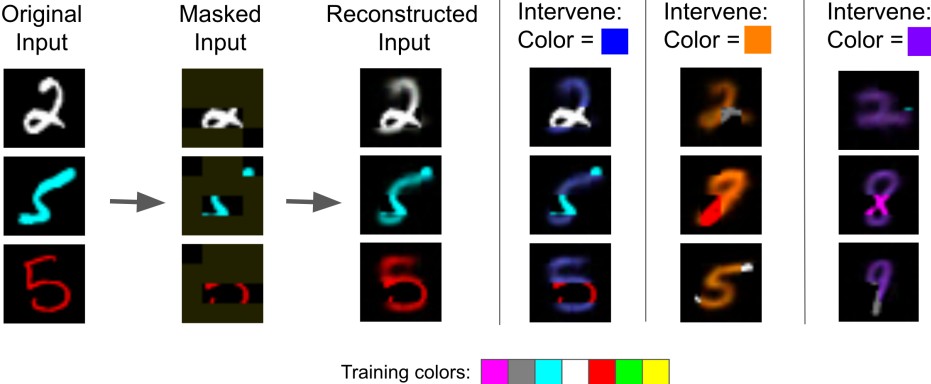

Figure 25: Results of our generative CBM on color MNIST. We can accurately generate digits in colors completely unseen during training. This highlights that CBMs are able to generate novel outputs.

# D  INTERPRETABILITY AND DEBUGGING

## D.1  FEATURE ATTRIBUTION AND COORDINATE SELECTION

The goal of our control experiments is to make small modifications to protein sequences that increase or decrease the value of a certain concept. While during training we randomly select which tokens to mask and predict, this is not an effective strategy for interventions. Instead we wish to identify which amino acids are increasing/decreasing a certain concept the most, and only mask and predict those tokens.

For example if we wish to increase Aromaticity of a protein, we wish to identify the amino acids which contribute the most to decreasing the Aromaticity, and intervene on those tokens. The most straightforward and principled method to measure this is occlusion, where we simply remove a each input token and replace it with a reference value one at a time, and see how our predicted concept changes for each. Since our model was trained with masking, the natural reference value is the mask token, so when removing a certain input we simply replace it with the mask token. Let $T \in \mathbb{R}^{v \times d}$ be the learned token embeddings where $v$ is the vocabulary size and $d$ is transformer hidden dimension. Then let $f_T(x) = [T_{x_0}, T_{x_1}, \ldots, T_{x_s}]$. The occlusion attribution for token $t$ of input $x$ on concept $i$ is then:

OCCLUSION:

$$A(x, i, t) = \hat{c}_i(f_T(x)) - \hat{c}_i(f_T(x_{x_t \leftarrow [MASK]}))  \tag{2}$$

Where $x_{x_t \leftarrow [MASK]}$ represents $x$ with the $t$-th token replaced by the mask token, and $\hat{c}_i(e(x))$ is the concept value predicted by our CBM model for concept $i$. concept prediction for concept $i$ on input $x$. For positive interventions, we typically then mask the 5% of the tokens (excluding padding) with the smallest attribution value(token being present reduces concept value), while for negative interventions we mask 5% of tokens with the largest attribution value(token being present increases concept value).

While this is a good method, it requires a separate forward pass for every individual token in the input, making it slow to run in large scale. Instead, as is common with feature attributions, we use a gradient based approximation of the occlusion value for up to 512 times speedup compared to occlusion. However, since our inputs are discrete, we cannot directly calculate gradients with respect to them. Instead, we calculate gradients with respect to the learned token embeddings of the model, and sum over these. Then

GRADIENT x (INPUT - MASK):

$$A(x,i,t) = \nabla_{f_T(x)_t} \hat{c}_i(f_T(x)) \cdot (T_{x_t} - T_{[MASK]}) \tag{3}$$

Where $f_T(x)_t$ is the $t$-th element of $f_T(x)$. This method is essentially a first order Taylor Approximation of the occlusion metric defined above. Alternatively this method could be seen as a version of Integrated Gradients Sundararajan et al. (2017) with only one gradient step and the mask token as the reference value. To see whether this approximation is sufficiently accurate to identify good amino acids to intervene on, we compare it's performance against occlusion itself in table 3. We also compared against some common feature attribution baselines described below:

GRADIENT x INPUT: Shrikumar et al. (2017)

$$A(x,i,t) = \nabla_{f_T(x)_t} \hat{c}_i(f_T(x)) \cdot T_{x_t} \tag{4}$$

GRADIENT: Simonyan et al. (2013a)

$$A(x,i,t) = ||\nabla_{f_T(x)_t} \hat{c}_i(f_T(x))||_1 \tag{5}$$

We compared all these methods, as well as a baseline of randomly choosing tokens to mask. We took a random sample of 50 proteins from the validation dataset, and intervened to increase/decrease each of the 14 biopython concepts one at a time, reporting the average change across concepts when masking 5% of the inputs in table 3. We can see choosing tokens to mask based on occlusion attribution performs the best as expected, however our first order approximation Gradient x (Input - Mask) also performed very well, with average intervention effect only 5% below using occlusion, while being around $100\times$ faster to calculate. On the other hand, simply looking at the magnitude of the gradient performed poorly, even worse than random selection, likely because it doesn't take into account whether said input is increasing or decreasing the concept value. Overall we can also see that choosing which tokens to mask is very important for controlled generation, and that with good attribution our interventions are more than twice as effective as randomly selecting tokens to intervene on.

| Method | Positive | Negative | Average |
|---|---|---|---|
| Random | 0.0312 | -0.0125 | 0.0218 |
| Gradient | 0.0173 | -0.0201 | 0.0187 |
| Gradient x Input | 0.0358 | -0.0269 | 0.0313 |
| Gradient x (Input - Mask) | 0.0480 | -0.0390 | 0.0435 |
| Occlusion | **0.0499** | **-0.0419** | **0.0459** |

Table 3: Average concept change after interventions using different feature attribution methods on our 24M-CBM model. The average column reports the average change in the desired direction.

Based on these results, for all other experiments in the paper we use the Gradient x (Input - Mask) method for our CB-pLM and CC-pLM, while the C-pLM uses random attribution as it does not make concept value prediction $\hat{c}$ which is needed for all other attribution methods.

### D.2 UNDERSTANDING CONCEPT/OUTPUT RELATIONSHIPS

To understand the model's ability to infer biophysical properties, we visualized the weights from the final layer for each concept in relation to each amino acid. Our findings indicate that the model successfully learns several key biophysical relationships as defined in the BioPython library (Figures 26, 27 and 28).

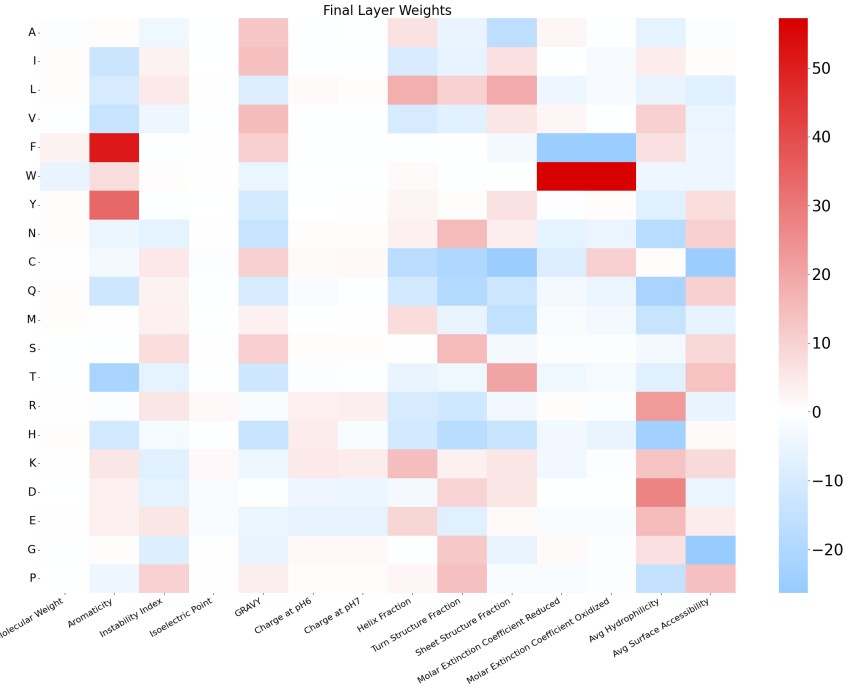

Figure 26: Visualization of the final layer weights in CB-pLM 24M.

**Some key insights include the following:**

- For both charge at pH 6 and pH 7, acidic amino acids (D, E) have negative weights, while basic amino acids (R, K) have positive weights. The difference in charge between pH 6 and pH 7 reflects the biophysical properties of H , as it contains an imidazole side chain with a pKa of 6.0 and contributes more towards positive charge at lower pH (Figures 27a, 27b).

- GRAVY, which defines hydropathy based on the Kyte-Doolittle scale (Kyte & Doolittle, 1982), is consistent with positive weights assigned to A, I, V, F, C, and M amino acids (Figure 27c ).

- Aromatic amino acids F, Y, and W have high weights for the aromaticity concept (Figure 27d).

- Average hydrophilicity uses the Hopp-Wood scale, which differs from the Kyte-Doolittle scale in its empirical definitions, assigns higher values to R, D, E, and K (Hopp & Woods (1981)). Although the scale also assigns lower values to aromatic residues, such as F, W, Y, we find that our model assigns lower weights for other amino acids which could reflect biases in the dataset (Figure 27e).

- Average surface accessibility is defined by the Emini Surface fractional probability (Cock et al., 2009), which is a lookup table describing the likelihood that a residue appears on the protein surface. The scale defines a low value especially for C and for other non-polar amino acids, though to a lesser extent (Figure 27f).

- Helix fraction, sheet structure fraction, and turn structure fraction are related concepts, are defined by fractions of I, L, F, W, Y for helix, A, L, M, E for sheet and N, S, G, P. While the weights do not closely match that of the BioPython definitions, we note that the weights are nonetheless

consistent with amino acids that often have special propensity to be part of helical structures, such as A, M, L, E, K (Leiro et al., 2017) (Figures 28a, 28b and 28c).

- Isoelectric point is a property closely associated with charge at pH 7, and likewise, we observe low values for D and E, and high values for R and K (Figure 28d).

- Molar extinction coefficient, which quantifies absorption of a protein at 280 nm, emphasizes the contributions of W. Likewise, the reduced vs oxidized terms of the molar extinction parameter differ only in how the oxidation state of C is considered, which is also visible in our model weights (Figures 28f, and 28e).

- Instability index is defined by a look-up table based on the dideptide decomposition of the sequence (Guruprasad et al., 1990). We observe that the model attributes high values to P, which often introduces kinks in the protein structure and disrupts secondary structures (Figure 28g).

- We note that the model weights does not always reflect the concept definitions. For example, W is the heaviest amino acid but is assigned a negative value, which likely reflects biases in the dataset. W is typically one of the least common amino acids, and relative frequency may vary by sequence length. However, the model correctly identifies F as contributing to high molecular weight, and A, and G towards contributing to low molecular weight (Figure 28h).

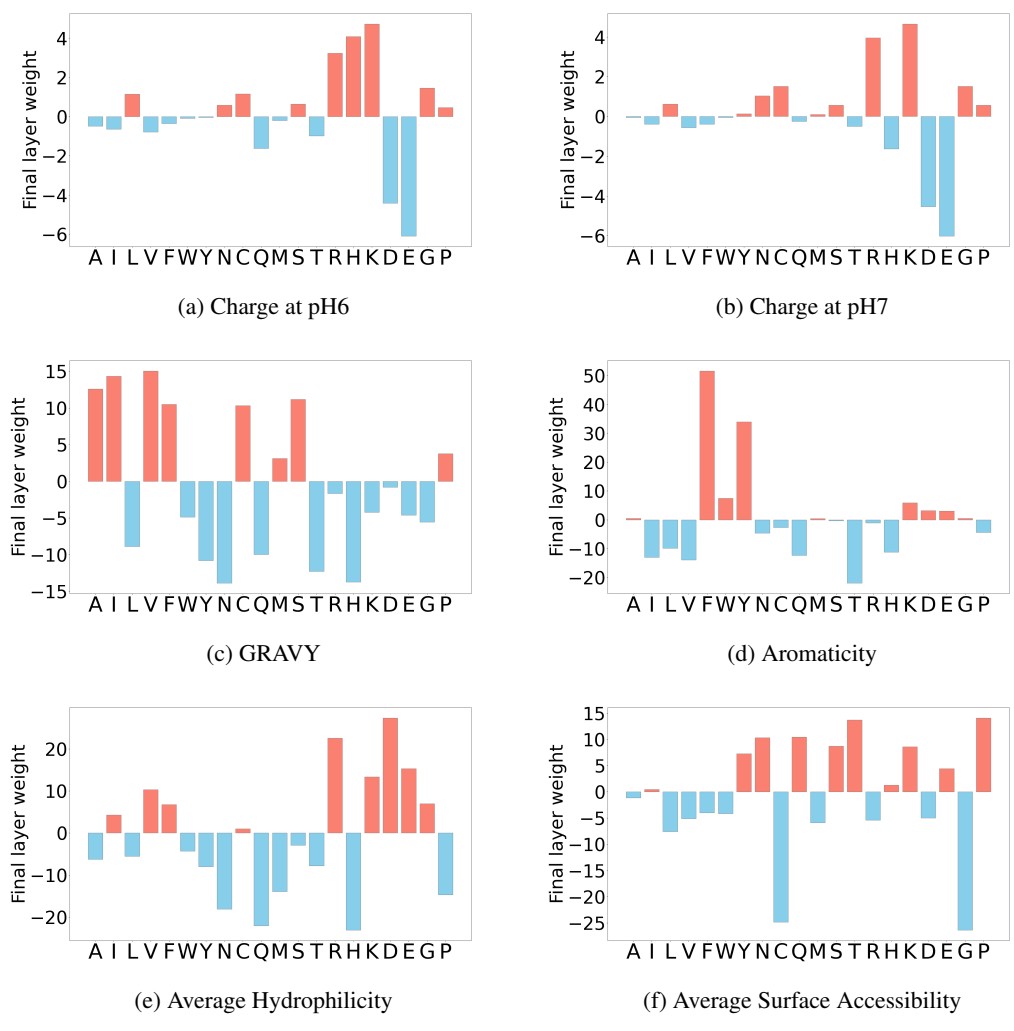

(a) Charge at pH6

(b) Charge at pH7

(c) GRAVY

(d) Aromaticity

(e) Average Hydrophilicity

(f) Average Surface Accessibility

Figure 27: The weights of concepts to amino acids in the last layer CB-pLM 24M.

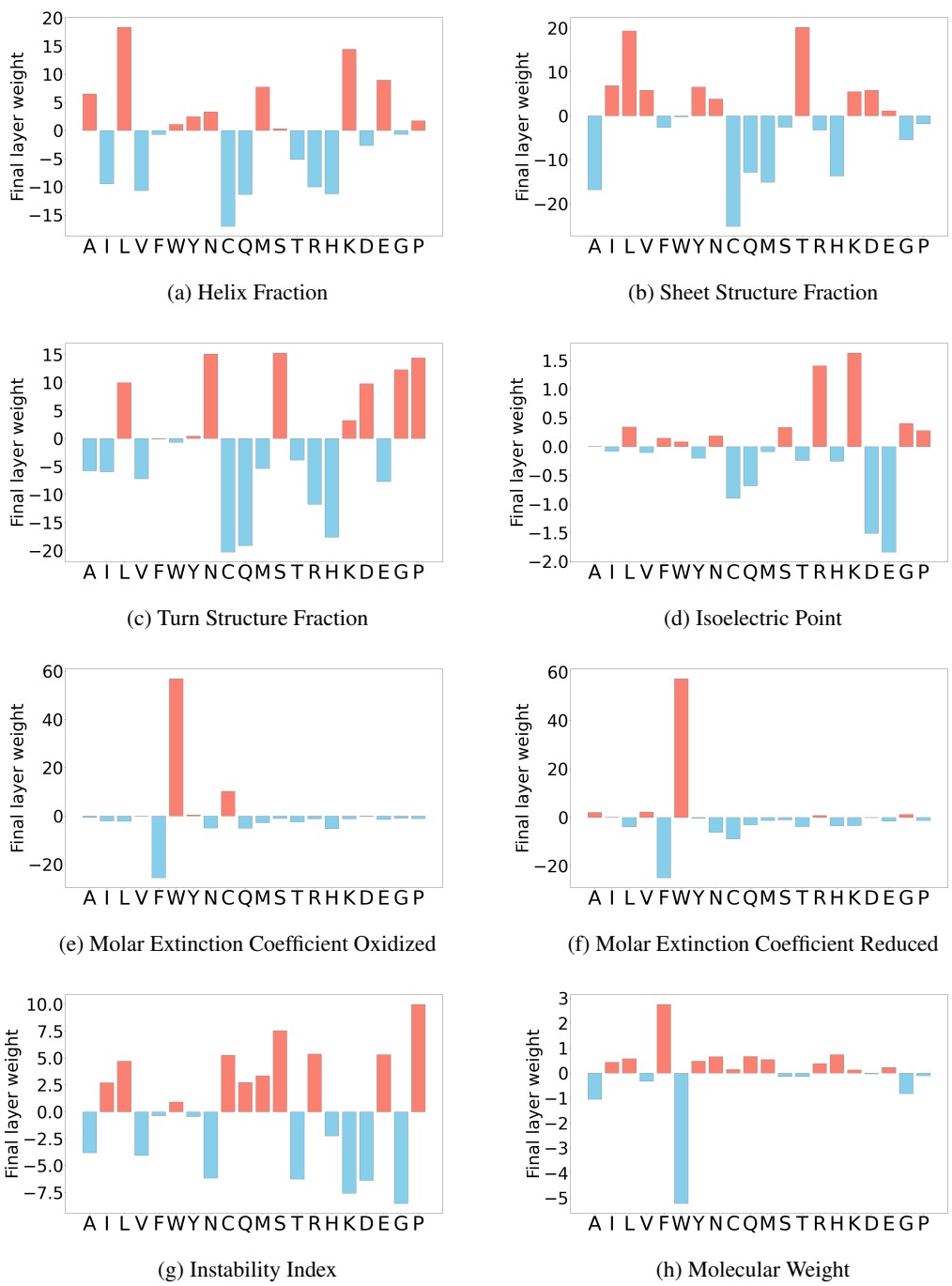

Figure 28: The weights for selected concepts to amino acids in the last layer CB-pLM 24M.

# E    EXTENDED RELATED WORKS AND DISCUSSION

## E.1    RELATED WORK

### LANGUAGE MODELS

Language models are increasingly widespread in their usage, spanning from natural language processing (NLP) (Achiam et al., 2023; Touvron et al., 2023; Team et al., 2023) to specialized fields such as chemistry (Bran et al., 2023) and biology (Cui et al., 2024; Lin et al., 2023). These models have demonstrated exceptional performance on intricate tasks, facilitating new frontiers in diverse applications (Brown et al., 2020; Hayes et al., 2024). However, the lack of concrete and effective interpretability methods limits their use in many high-stakes applications, engendering distrust among domain experts, and raising concerns regarding regulatory compliance, safety, and alignment (Goodman & Flaxman, 2017; Gabriel, 2020).

Current interpretability research for large language models (LLMs) focuses on pre-trained models, either by explaining the model's predictions (Zhao et al., 2024) or by analyzing the internal circuits of the network through mechanistic interpretability (Räuker et al., 2023; Bricken et al., 2023). Recent work in mechanistic interpretability (Bricken et al., 2023) demonstrated that sparse autoencoders (SAEs) could uncover interpretable monosemantic features. To apply this on large-scale production models, it involves training very large SAEs, retrieving millions of features, and then trying to assign meaning to these features using an autointerpretability approach (Bills et al., 2023).

Although there have been promising results in controlling large production models like Claude using such SAEs, it remains unclear how this approach will help when controlling for a concept not found in the SAE or when autointerpretability is not feasible, such as in domains like proteins. Concept bottleneck language model (CB-LM) demonstrates similar capabilities (see Appendix C.3.1 for experiments). However, for CB-LMs, there is no need to train an additional SAE and conduct a post-hoc search for the meaning of different concepts, as each concept's encoding is known by design.

### CONCEPT BOTTLENECK MODELS

Concept Bottleneck Models (CBMs) (Koh et al., 2020) incorporate interpretability into neural networks by inserting an interpretable "concept bottleneck" layer into the network and mapping the inputs to a set of human-understandable concepts, which are then used to make the final prediction.

Recently, Ismail et al. (2023) demonstrated that a concept layer can be integrated into generative models (CBGMs). Intervening on this layer allows for controllable generation, and examining the activations provides concept-level explanations. Their work focused on image generation and evaluated the effectiveness of this approach in architectures where the entire input is represented as an embedding, such as GANs, diffusion models, and VAEs.

In this paper, we extend this approach to language models, demonstrating that we can achieve global control over the entire input while retaining token-level generation. To make this adaptation, we inserted a concept bottleneck layer into a language model and created a novel concept module. This differs from CBGMs in several key aspects: (a) We focus on sequential data where inputs are represented as tokens. (b) We employ a unique model architecture tailored to language models. (c) We introduce a new intervention procedure specific to our architecture.

A few very recent works have proposed CBMs for text classification tasks Tan et al. (2024); Sun et al. (2024), but this is a much easier and less interesting setting than generative language modeling which we address, as the input and output space of generative models is much larger and cannot be adequately modeled using only known concepts, highlighting the need for an unknown part in the embeddings.

Recent work Yuksekgonul et al. (2022); Oikarinen et al. (2023); Yang et al. (2023); Rao et al. (2024) has also proposed methods to train Concept Bottleneck Models for image classification that do not require labeled concept data by leveraging pretrained multimodal models. While interesting, this approach reduces reliability of concept predictions, and is not feasible in protein setting as no good pretrained general concept predicting models are available.

Similar to ours, some previous work has proposed including an unknown part in the embeddings to improve task performance in the image classification settings Yuksekgonul et al. (2022); Sawada & Nakamura (2022), but they employ quite different methods to achieve this in a different setting.

## PROTEIN LANGUAGE MODELS

Protein language models are widely used in biological machine-learning research, trained on extensive protein sequences spanning the evolutionary tree of life. Although pLMs have numerous critical applications in healthcare and drug discovery (Hie et al., 2024), they currently lack interpretability. Their performance is often explained through intuitive arguments about compression and the learning of co-evolutionary patterns (Rives et al., 2019; Lin et al., 2022; Hayes et al., 2024), but it remains unclear whether these models truly capture meaningful protein concepts.

Historically, pLM architectures, loss functions, and training setups have closely mirrored the original masked language model setup (Devlin et al., 2018), with the primary differences being the vocabulary and dataset. Our goal is to develop a pLM that offers controllability, interpretability, and debuggability, making it more reliable for design applications. While some protein language models support conditional generation by concatenating different protein functions and properties to the inputs (Shuai et al., 2021; Madani et al., 2020; Hayes et al., 2024), in contrast to our work, they do not provide any mechanisms for interpretability, debuggability, or offer any insights on what a model has learned.

### E.2 DISCUSSION

#### CB-LM FOR OTHER DOMAINS AND ARCHITECTURES

In this paper, we applied our proposed architecture to masked language models for proteins. This versatile architecture can be adapted to any domain, such as NLP, by simply changing the training dataset, vocabulary, and concepts while keeping the model itself unchanged. Additionally, it can be easily adapted to other types of language models, such as autoregressive models, as demonstrated in Figure 29. For autoregressive models, the embedding output from the transformer, typically used to predict the next token, is passed to the CB-layer to extract the known concept embedding and to an orthogonality network to extract the unknown concept embedding. These two embeddings are then concatenated and used to predict the next token.

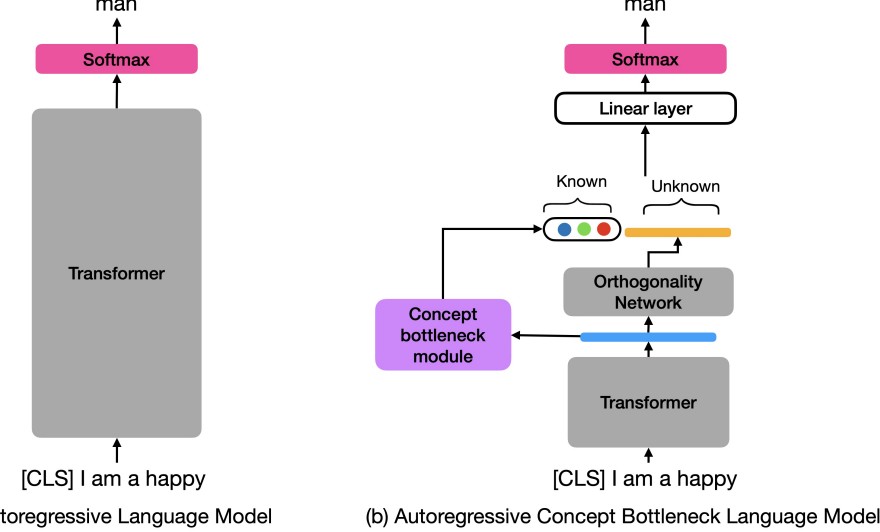

Figure 29: Inserting concept bottleneck into an autoregressive language model.

ADDING NEW CONCEPTS TO A TRAINED CB-LM

An intuitive question that might arise is how we can control for new concepts. Specifically, if we are given a dataset with a new concept, can we fine-tune CB-LM to incorporate it? This can be achieved by creating a new CB-LM with an additional concept and copying the weights from the original model. In this setup, the only randomly initialized weights are for the new concept embedding $e_{k+1}$, the concept classifier head for the new concept, and the linear decoder. The new model can then be trained to match the old model's concept output for existing concepts while aligning with the ground truth for the newly added concept.

GENERALIZATION TO NEW CONCEPT COMBINATION

In many real-world applications, we may need to generate sequences with previously unseen concepts or novel combinations of concepts not present in the training dataset. CB-LM provides this capability, as long as the new concepts can be expressed as functions of the existing ones. We demonstrate this capability in a toy task detailed in Appendix C.3.2, where we train a masked autoencoder vision transformer (MAE-ViT) (He et al., 2022) with our architecture on a simple color-MNIST generation task. Our results show that the architecture can successfully generate blue-colored MNIST digits, despite the model never having encountered the color blue in the training dataset.

