# OpenReview forum: "Concept Bottleneck Language Models For Protein Design"
_ICLR.cc/2025/Conference — ICLR 2025 Poster_

### Official Review · Reviewer_AGed · 2024-11-04

**Soundness:** 4
**Presentation:** 4
**Contribution:** 4
**Rating:** 8
**Confidence:** 3

**Summary:**

This paper takes the approach called concept bottleneck language modeling and it applies it in the protein language model context.  This provides creative control in terms of human-interpretable biological concepts, as well as the ability to debug models.  The proposed approach appears to work better than other generic approaches to controllable generation such as ProGen/CTRL, as demonstrated in experimental comparison with other models trained by the researchers themselves.  A case study with redesign of siltuximab shows the ability to quite precisely control certain properties, though no wet lab experiments are shown to verify.

**Strengths:**

Human-interpretable creative control is a super key in generative AI broadly, but especially in protein design.  As such, the proposed approach based on the concept bottleneck is quite compelling.  The computational experiments provide evidence that the proposed architecture is effective.  The paper is written in a clear way.

**Weaknesses:**

No wet lab experiments were done to verify results.  This would strengthen the paper considerably.

Comparisons to other architectural approaches are based on the authors' own implementations, rather than by external research groups that may be motivated to optimize the other architectures more.  Notwithstanding, the performance differences are quite considerable.

No discussion on biosecurity risks is given.

**Questions:**

If even a small wet lab experiment is possible, I would be super curious to see the result.

---

> ### Author Response · Authors · 2024-11-22
> **Response to Reviewer AGed**
>
> Thank you for your positive feedback, we are glad you found our approach compelling. Below, we address your concerns point-by-point.
>
> - **Re: No wet lab experiments were done to verify results. This would strengthen the paper considerably.**
>   Please find our response in the general response **R2**.
>
> - **Re: Comparisons to other architectural approaches are based on the authors' own implementations**
>   We have implemented baselines ourselves because we are unaware of open-source models that offer control for the same concepts we are testing.
>
> - **Re: No discussion on biosecurity risks is given.**
>   Thank you for pointing this out; we have included biosecurity risks in the discussion section in the updated manuscript.

---

> > ### Author Response · Authors · 2024-11-25
> > **Any follow up questions**
> >
> > Dear Reviewer  AGed,
> >
> > We thank you again for the positive feedback on our work, we really appreciate it.  Given that the end of the discussion period is approaching, we would like to ask if you have any further concerns or questions.
> >
> > Thank you again!

---

### Official Review · Reviewer_i9uk · 2024-11-04

**Soundness:** 3
**Presentation:** 3
**Contribution:** 4
**Rating:** 8
**Confidence:** 3

**Summary:**

This work applies concept bottleneck models to protein language models. The approach uses fairly standard ideas in concept bottleneck models, but they are well-executed and lead to novel applications in e.g. controllability.

**Strengths:**

The controllability experiments are fairly convincing, though the stated application on debugging is fairly cursory and could be improved (the intervention to test debugging is fairly blunt, and also the results are just hand inspection and thus not reproducible). However, for me the controllability experiments are enough to be a solid contribution.

The perplexity results were moderately convincing, though more discussion of comparisons to LBSTER and ESM2 would be useful -- it was not clear to me for instance if your method was given extra information (the tags) that LBSTER / ESM2 did not have. It was also strange that the largest model had worse perplexity.

The writing is overall good and easy to follow, but I felt that the level of detail in some experiment descriptions was not enough to be fully reproducible from the text.

**Weaknesses:**

See above (strengths and weaknesses are merged)

**Questions:**

There are some minor typos e.g. "00:01 01:15" in Section 3.1, as well as a missing line break before \textbf{Scaling} in the same subsection.

---

> ### Author Response · Authors · 2024-11-22
> **Response to Reviewer i9uk**
>
> Thank you for your positive feedback, we are glad you found our work well executed and having novel applications. Below, we address concerns point-by-point.
>
> - **Re: Perplexity**
>   Our model, CB-pLM, does not require tags during inference. Reviewer ey43 pointed out that our validation dataset might have been included in the training dataset for LBSTER/ESM2. To address this, we repeated the experiments using the Mason dataset [1] and updated the manuscript with the new values. We have also clarified which models require tags and which do not. Regarding the perplexity results, we did not observe significant performance gains beyond the 650M model, noting that our 650M model already outperforms ESM2 3B. Please find our reasoning for this observation in the general response **R1**.
>
> - **Re: Typos**
>   Thank you for catching this. We have corrected this in the updated manuscript.
>
> ### References
>
> [1] Mason, D. M., et al. "Optimization of therapeutic antibodies by predicting antigen specificity from antibody sequence via deep learning."

---

> > ### Author Response · Authors · 2024-11-25
> > **Any follow up questions**
> >
> > Dear Reviewer  i9uk,
> >
> > We thank you again for the positive feedback on our work, we really appreciate it.  Given that the end of the discussion period is approaching, we would like to ask if you have any further concerns or questions.
> >
> > Thank you again!

---

### Official Review · Reviewer_e4sF · 2024-11-08

**Soundness:** 2
**Presentation:** 1
**Contribution:** 2
**Rating:** 5
**Confidence:** 2

**Summary:**

The paper introduces a novel protein language model designed to provide predictions that can be interpreted by human experts without compromising performance. The proposed model, dubbed Concept Bottleneck Protein Language Model (CB-pLM), empirically proves to be easier to control and interpret over previously proposed State of the Art Protein Language Models.

**Strengths:**

Developing and testing new models for protein modeling is an important problem. In particular, having methods that can be interpretable and that can allow better control for drug design is a high impact project.

**Weaknesses:**

The paper contains many typos, it is meandering and unfocused, and generally is very hard to read (with many repetitions and vaguely defined concepts).

For example, what does “debuggable model“ mean? This is stated as one of the main contributions of the proposed architecture, hence the reader expects it to be clearly defined and thoroughly assessed through empirical validation. In the current version of the manuscript this is not properly defined and hence not convincingly validated with experiments.

What are the novel contributions specific to the Protein Language domain? Can the authors separate the specific novelty introduced for their protein language model? What are the main differences with the previously proposed concept Bottleneck Generative Models? If the main contribution of the paper is to test whether CB-LMs work on the Protein domain then it would be good to devote more space in the paper to the empirical validation with a thorough comparison with other State of the Art models at different scales.

The discussion section briefly mentions some properties of the model that have not been tested in the main empirical section of the paper. It would be good either move the strong empirical evidence supporting concept composition and generalization to novel concepts (added with fine-tuning) in the main or, if results are not convincing enough, remove the claim. Similarly, in the conclusions (and abstract) it is mentioned that the current model has been tested up to the 3B scale and compared with other public models. However I did not manage to find evidence of experiments on scales larger than 150M (see for example Table 1 in which the proposed method does not perform comparably with others).

**Questions:**

1. Line 76, what are “human understandable concepts”? This is not very well defined in the paper and it is left vague. Similarly, what does debuggable model mean? Even if the proposed model leverages a bottleneck to force some representations to align to a set of pre-specified “concepts” the whole model before the bottleneck is a complete black-box. Does debuggable mean that the user can inspect the black box before the bottleneck too?
2. In the abstract and intro it is mentioned that the proposed model has been scaled to 3B parameters, however there are no empirical results showcasing models larger than 150M (most of the results are reported on the small 24M model). Furthermore, in Table 1 the comparison with other State of The Art models suggests that the proposed model regresses in perplexity over similarly sized non-interpretable models. Can the authors provide an apples to apples comparison with other public models at scales larger than 150M?
3. Is the current method supposed to scale to a large number of protein sequences? Will it generalize in the case of mislabelled or unlabelled samples? Given annotation cost, a model capable of leveraging partially annotated/unlabeled samples is more likely to scale.
4. What is the maximum context length (length of the protein sequence) that the model supports?
5. It is not clear how the concepts in the bottleneck layer should be defined and why the model needs to learn to associate specific concepts to “disentangled” representations in one-to-one relation with the concepts bank. From the paper it appears that the concepts are annotated and hence the “factor of variations” are known at training time. This could become problematic since scaling on annotated data is not always possible. Does the proposed method allow one to not directly observe the factors of variations and still learn some minimal and sufficient concept bank, similarly to the method proposed in [1]? Furthermore, when annotations are used, are the learned concepts compositional?
6. Experiments in section 4.1.1 and 4.1.2 have been conducted on ablations of the proposed method. Why is it not compared with other State of The Art methods of similar sizes?


[1] M. Fumero et al., “Leveraging sparse and shared feature activations for disentangled representation learning”, NeurIPs 2023.

Minor:
- Line 121, typo.
- Line 125, labels in match with Figure 2 even if they are not related.
- Line 226 repeated.
- Line 245, typo

---

> ### Author Response · Authors · 2024-11-22
> **Response to Reviewer e4sF part 1**
>
> Thank you for your review. Below, we address the weaknesses and respond to your questions point-by-point.
>
> ### Re: Weaknesses
>
> - **W1: Paper contains many typos, it is meandering and unfocused**
>   We have addressed the specific points in our revised manuscript which is hopefully easier to follow. If you have additional suggestions on parts you still find unclear please let us know.
>
> - **W2: What does “debuggable model“ mean?**
>   Please find our response in the general response **R3**.
>
> - **W2: What are the novel contributions?**
>   Our work introduces the first concept-bottleneck language model. The original concept bottleneck generative model (CBGM) paper focused on diffusion models, VAEs, and GANs, which have inherently different architectures and properties compared to language models. Adapting CBGMs to language models is not straightforward. The main difference is that in language models, generation is done tokenwise while concepts are defined over the entire sequence, while for diffusion, VAE, and GANs, the entire output is generated at once. To adapt concept bottleneck to language models, we have made the following changes:
>   - **Architecture:**
>     - In CB-LMs, the concepts are extracted from the CLS token; in CBGMs, the concepts are extracted from the entire representation.
>     - In CB-LM, we add an orthogonality network to remove concept information for every token. This was not done in CBGMs since it was not needed.
>     - CBGMs use a CEM for the CB-layer i.e., for each concept, there is a network that predicts a concept embedding from the input representation; this did not scale well for LLMs. So, in CB-LMs, for each concept, we learned a concept representation (this representation is learned but is not dependent on the input as in CEMs); in CB-LMs, the concept embedding is the concept representation multiplied by the concept value as described in section 2.1. We find this significantly improves control and reduces the number of added parameters.
>   - **Training:**
>     - We added noise regularization to the input embedding to help with faithful attribution as this is a requirement for successful interventions in our setup and again not in CBGMs.
>   - **Interventions:**
>     - We introduced an intervention scheme specific to language models, again this is not present in CBGMs, the intervention scheme is described in Section 2.3.
>
>   Overall, the novelty of our work is not just in the domain but rather the architecture, training scheme, and intervention scheme as well as the domain. Regarding empirical validation, we compared our model against state-of-the-art open-source protein language models in terms of perplexity. However, our model offers unique control, interpretability, and debuggability capabilities that current protein language models do not, making direct benchmarking in these aspects impossible.
>
> - **W3: The discussion section briefly mentions some properties of the model that have not been tested in the main empirical section of the paper.**
>   Thank you for pointing this out, we have updated this section in the manuscript to be titled "discussion and future work” and rewrote the section.
>
> - **W4: Scale experiments**
>   The results for the 650M and 3B models are in Appendix C. The reason they are in the appendix and not in the main text is that (a) we do not have 650M and 3B baseline models to compare with, and it will be too expensive and non-environmental-friendly to train baselines of this size. (b) we did not observe much gains in performance with scaling (Please find our response in the general response **R1**).
>   We trained these size models in the first place to show this architecture can, in fact, be scaled up easily. We have repeated the perplexity experiments to address concerns raised by ey43 and added 650M and 3B ESM2 baselines to address your concerns. Our model significantly outperforms ESM2 in different scales.

---

> ### Author Response · Authors · 2024-11-22
> **Response to Reviewer e4sF part 2**
>
> ### Re: Questions
>
> - **Q1: What are “human understandable concepts”? What does debuggable model mean?**
>   A human understandable concept is an input feature or transformation of that input feature that a human (typically a domain expert) can attach meaning to. For debuggable model, please find our response in the general response **R3**.
>
> - **Q2: Can the authors provide an apples to apples comparison with other public models at scales larger than 150M?**
>   Please see **W4**.
>
> - **Q3: Is the current method supposed to scale to a large number of protein sequences?**
>   Since CB-pLM can scale to a large number of parameters, we do not foresee any issues with supporting more training data. Complete annotations for all protein sequences are not required; a subset of annotated data is often sufficient to learn the desired concepts. In fact, during training, we combined two datasets, UniRef50 and Swissport; the UniRef50 dataset lacks concept annotations found in the Swissport dataset, resulting in less than 3% of the data being fully annotated. Details on how we handle missing concepts can be found in Appendix A.1.
>
> - **Q4: Is the current method supposed to scale to a large number of protein sequences?**
>   The maximum context length is 512, similar to [1]. Note that a typical antibody is 249 amino acids. See model details in Appendix B.2.
>
> - **Q5: It is not clear how the concepts in the bottleneck layer should be defined.**
>   Very good questions! We’ll address these in parts.
>   - **Defining concepts:** The focus of our work is scientific applications where there is often ample domain expertise and prior knowledge about the underlying factors of variation. However, the exact functional relationship between such factors and the output might be unknown—hence why machine learning is being applied. One such setting is drug discovery where certain key biophysical properties are known, and have been experimentally confirmed, to affect a protein’s function. Given such knowledge, domain experts often want to assess the extent to which a generative model’s representations encode these properties. We’ll address your point about unknown concepts soon, but we contend that, in scientific applications, defining concepts amounts to encoding these factors as concepts in the model. Across various applications, we appeal to domain expertise to define concepts.
>   - **Why should the model learn to associate specific concepts to “disentangled" representations:** To start, one of our key constraints is: we enforce orthogonality between the known and unknown concepts, but not ‘between’/’among’ known concepts. As we previously discussed, we rely on a domain expert to specify known concepts. Consequently, we are relying on expert knowledge prior to prune the concept set to a minimally sufficient list that consists of factors that can be expected to be disentangled. Further, our proposal includes a loss function that requires the unknown concepts to be orthogonal to the unknown ones. A key point here is that if the list of known concepts is not disentangled, then we cannot expect the representations learned by the CB layer to also be disentangled. The key constraint we enforce is orthogonality between known concept and unknown concept sets. We expect future work to address limitations of our current proposal.
>   - **Scaling Concept Annotation:** Complete annotations for the entire dataset are not required; a subset of annotated data is often sufficient to learn the desired concepts. Details on how we handle missing concepts can be found in Appendix A.1.
>   - **Unsupervised Concept Discovery:** Without any concept labels, domain specific assumptions, and/or constraints to the learning process, automatically learning the factors of variation—unsupervised—from data alone is not identifiable [2]. Another key challenge is that without supervision or constraints, the unsupervised concepts that a model learns might be difficult to interpret for humans and domain experts. Despite these challenges, unsupervised concept discovery with concept bottleneck layers is feasible, and can be directly applied here as well.
>   - **Compositionality:** We do not explicitly address compositionality beyond assumptions that go into the domain expert’s concept selection process. Our proposal can be extended in straightforward ways to incorporate concept compositionality [3].
>
>   We thank the reviewer for these important questions, and hope our answers help contextualize the scope of our contributions.
>
> - **Q6: Experiments in section 4.1.1 and 4.1.2 are not compared with other State of The Art methods of similar sizes?**
>   CB-pLM offers novel capabilities not available in current open source pLMs, i.e., SOTA pLMs do not offer control over the concepts we are testing. Please let us know if you have any particular models in mind.

---

> ### Author Response · Authors · 2024-11-22
> **Response to Reviewer e4sF part 3**
>
> ### Re: Typos
> Thank you for pointing this out. We have corrected the typos in lines 121 and 245. Regarding line 226, the repetition is intentional to emphasize and clarify the novel capabilities offered by CB-pLM.
>
>
>
> ### References
>
> [1] Frey et al. "Cramming Protein Language Model Training in 24 GPU Hours."
>
> [2] Locatello et al. "Challenging common assumptions in the unsupervised learning of disentangled representations."
>
> [3] Stein et al. "Towards compositionality in concept learning."

---

> ### Author Response · Authors · 2024-11-25
> **Any follow up questions**
>
> Dear Reviewer e4sF,
>
> Given that the end of the discussion period is approaching, we would like to ask if you have any further concerns or questions, particularly as a follow-up to our response?  If we have addressed all your concerns, we hope you will consider increasing your score.
>
> Thank you in advance!

---

> > ### Comment · Reviewer_e4sF · 2024-11-26
> >
> > Thank you for your thorough and clear responses! Most of my concerns have been addressed, and I appreciate the authors' willingness to revise parts of the previous manuscript. I am raising my score as a result.

---

### Official Review · Reviewer_QrUQ · 2024-11-09

**Soundness:** 2
**Presentation:** 3
**Contribution:** 2
**Rating:** 6
**Confidence:** 2

**Summary:**

This paper proposes a Transformer Masked Language Model (MLM) variant of concept-bottleneck generative models (CBGM) with three goals in mind: controllable generation, interpretability, and debuggability. This paper adapts the training objectives of CBGM to the masked language model architecture, and proposes a control method specifically for their CB MLM. Evaluation was conducted on the domain of protein sequence modeling, for controllability, interpretability and debuggability. The proposed model leads to improvements over non concept-bottlenecked baselines on controllability. The proposed model demonstrates interpretability and debuggability via inspection of the weights of the linear decoder head.

Comment: I have experience with language modeling but not with protein domain, so my review will focus more on the method part of the paper (as recommended by AC). Please take this into account when considering my review and recommendation.

**Strengths:**

* Adapting concept-bottleneck generative models to masked language models is an interesting and novel idea.
* Using integrated gradients to find tokens to resample to is an interesting and novel idea.
* Writing is generally easy to follow.
* This work focuses on improving controllability, interpretability, and debuggability of language models (and protein sequence in particular as an example) which is an important topic with large potential impact.

**Weaknesses:**

Method

(considering a broader application of the proposed method to general language/sequence modeling, as authors mentioned in the abstract)

M.1 The model proposed is only a conditional generative model given a partial sequence rather than a full generative model of an entire sequence. Classical masked language models (like BERT) do not model the joint distribution over sequences [1]. There are ways to derive joints from MLMs by making their conditionals consistent with one another (see, e.g. [1] and the related works cited there), or by modifying the training objective so that the conditionals are encouraged to be consistent during training (e.g. [2] [3]). Other protein models based on MLM (e.g. [3], which the authors cited) have done the latter kind of training modifications to enable sampling full sequences. I'm not sure how this limitation matters for protein modeling, could authors give more discussions about this?

M.2 Naturalness of the controllably generated sequence is not considered during intervention, but only used as a measurement afterwards. See point below for consequence.

M.3 The proposed controllability technique only proposes changes local to a particular known sequence x, by changing n% (5 in experiments) of x at a time. Again, I cannot comment on the significance of this limitation for protein modeling. For other modalities like images and language, it can be difficult to change sequence-level concepts via chaining a sequence of such local changes. Even if it is possible, since naturalness is not considered as a factor during intervention but only afterwards, applying a sequence of interventions could lead to going outside of the training distribution (e.g. the interventions produced in Figure 20 goes outside of the color-MNIST training distribution, where all digits are uniformly colored). Empirically though, (Fig 4b 15 16) the proposed method seems to outperform baselines so maybe this is less of an issue in practice, for the concepts in this paper and/or for the protein domain.

Experiments

E.1 The model is trained with 25% masking, but is used to generate with 5% masking in the experiments. This could make the model worse because it’s a different input distribution. Is there a particular justification for masking 5%?

E.2 When evaluating controllability against baselines (and more specifically CC-pLM, since it's closer to CB-pLM), it’s not clear whether the gain comes from intervening at better positions (due to applying integrated gradients to a different architecture), better sensitivity of the model to interventions (due to the proposed concept bottleneck), or a combination, making it hard to evaluate the limitations / benefits of the architecture versus the intervention technique.

[1] Hennigen and Kim, 2023. Deriving Language Models from Masked Language Models.

[2] Germain et al., 2015. MADE: Masked Autoencoder for Distribution Estimation.

[3] Hayes et al., 2024. Simulating 500 million years of evolution with a language model.

**Questions:**

Q1 How does compute / training vary across C-pLM, CC-pLM, and CB-pLM. Is total compute/wallclock time controlled (e.g. 24 hr, as the cited cramming paper) or perhaps early stopping on validation is used?

Q2 It is interesting that there is negative scaling happening in figures 10 and 11, with sometimes large decreases in performance as the model becomes larger. Out of curiosity, do authors have any speculations about why this happens?

Q3 In lines 142-144, “This enables us to answer [… q1…], or [q2] counterfactual questions like if we want to decrease hydrophobicity, which amino acid should we replace “E” with?”. I understand q1 is an interpretability question and linear decoder does help with that. For q2 however, I am not sure I fully understood. Is it accurate to rephrase it as “if we intervened on the concept strength of hydrophobicity, which amino acid would the decoder output?” My confusion is on the verb “enables” – I believe the linear layer gives you a more interpretable answer to q2 (say compared to a nonlinear decoder), but you would still be able to answer q2 with a nonlinear decoder,  by intervening on the concept value $\hat{c}_i$ and then running a forward pass of the decoder.

Q4 What’s the spread along the x-axis on figure 4b? Is it naturalness? It might be helpful to also plot the reference lines for no-change (for both naturalness and intervened property) as you do in figure 4a.

---

> ### Author Response · Authors · 2024-11-22
> **Response to Reviewer QrUQ part 1**
>
> Thank you for the detailed review and feedback. We are pleased that you found our adaptation of concept bottleneck models to MLMs and the use of integrated gradients for resampling both interesting and novel. We are happy to clarify any design choices we made, especially those influenced by the specific requirements of the protein domain rather than NLP. Below, we address the weaknesses and respond to your questions point-by-point.
>
> ### Re: Weaknesses
>
> - **M.1: Generating full sequences from MLMs.**
>   Our approach can be extended to autoregressive models by making similar architectural changes, as described in Figure 27. For generating the full sequence using the current MLM, one can employ the approach described by ESM3 [1]; starting from a fully or partially masked context, tokens can be sampled one at a time or in parallel, in any order, until all positions are fully unmasked. However, many studies have noted that full sequence generation is not as useful or practical for antibody protein design and therapeutics [2,3,4]. Therefore, we do not focus on this aspect in the context of this paper.
>
> - **M.2: Naturalness of CB-pLM designs**
>   To address your concern regarding the naturalness of the proposed designs, we ran additional experiments to verify that our interventions do not compromise the integrity of the sequences. For the Siltuximab experiment, we folded designs from all models using ABodyBuilder2 [5] and calculated various protein surface properties using the Therapeutic Antibody Profiler (TAP) [6], which employs physics-based computations. Ideally, we aim for minimal changes from the original antibody. The results, plotted in Figure 20, show that CB-pLM designs are the least disruptive compared to other methods, where the dotted line in the figure represents the original antibody. The TAP metrics confirm that our CB-pLM designs maintain naturalness more effectively than other approaches.
>
> - **M.3: The proposed controllability technique only proposes changes locally.**
>   In protein design, the goal is generally to change a specific concept with minimal alterations to the sequence, as larger changes can introduce various liabilities or cause manufacturing difficulties. Prior work [2,3,4,7] has shown that significant changes to a protein's properties can be achieved with very few edits. Moreover, [8] has demonstrated that the functional effects of changes to an antibody sequence are not only non-additive but can exhibit state-dependent higher-order interactions, a phenomenon known as epistasis [9]. Overall, it is well established in the literature that even single-point mutations can have drastic effects on sequence-level concepts. Therefore, in our evaluation, we focused on making localized changes, which are well-suited for protein modeling and help maintain the naturalness of the sequences while achieving the desired modifications.
>   For the color MNIST, the experiment's goal is generating concepts outside the training distribution, showing the intervention generalizes successfully beyond the training distribution. This is unrelated to the masking percentage since, in this experiment, the percentage of masking in training and intervention is the same.
>
> - **E.1: The model is trained with 25% masking but is used to generate with 5% masking in the experiments.**
>   Please refer to our response to M3 for a detailed justification. To demonstrate that our model maintains consistent performance with different masking percentages, we repeated the intervention experiments with a 25% masking setup. The results, found in Figure 17 of the updated manuscript, show that the outcomes are consistent with those obtained using 5% masking. In both cases, CB-pLM successfully shifts the concept in the correct direction and significantly outperforms C-pLM and CC-pLM.
>
> - **E.2: Difficult to evaluate if performance gains are due to architecture changes or intervention technique.**
>   In Table 3 of Appendix D, we show that even when masking is applied randomly with the CB-pLM model, we achieve an average intervention effect of 0.0218. This performance is superior to the average change of 0.013 observed with CC-pLM using targeted masking. This demonstrates that CB-pLM is inherently more sensitive to interventions. Moreover, Table 3 illustrates that better localization, achieved through targeted interventions, further enhances our method, increasing the intervention magnitude by approximately 2x. This indicates that while the architecture of CB-pLM contributes significantly to its improved performance, the effectiveness of our intervention technique also plays a crucial role.

---

> > ### Author Response · Authors · 2024-11-22
> > **Response to Reviewer QrUQ part 2**
> >
> > ### Re: Questions
> >
> > - **Q1: How does compute / training vary across C-pLM, CC-pLM, and CB-pLM?**
> >   For a fair comparison, we set the maximum number of steps for all models to be the same 100K steps; models with the same size have the same batch size, but due to a few additional layers and losses added in our CB-pLM models take ~10% longer to train.
> >
> > - **Q2: Negative scaling**
> >   Please find our response in the general response **R1**.
> >
> > - **Q3: Lines 142-144**
> >   You are correct; thank you for pointing this out. We removed the word "enables" and replaced it with "gives us an interpretable answer" in the updated manuscript.
> >
> > - **Q4: What’s the spread along the x-axis on figure 4b?**
> >   In Figure 4b, we test the effect of iterative interventions to demonstrate that we can improve the property over multiple iterations. The x-axis represents the iteration number: iteration 1 is the intervention on the original sequence, iteration 2 is the intervention on the sequence generated in iteration 1, and so on. There is no naturalness plotted in this figure. As per your request, we have updated the y-axis to include a no-change reference line for better clarity.
> > -----------------
> > ### References
> >
> > [1] Hayes et al. "Simulating 500 million years of evolution with a language model."
> >
> > [2] Shuai, Ruffolo, and Gray. "IgLM: Infilling language modeling for antibody sequence design."
> >
> > [3] Frey et al. "Protein discovery with discrete walk-jump sampling."
> >
> > [4] Gruver et al. "Protein design with guided discrete diffusion."
> >
> > [5] Brennan Abanades et al. "ImmuneBuilder: Deep-Learning models for predicting the structures of immune proteins."
> >
> > [6] Raybould and Deane. "The therapeutic antibody profiler for computational developability assessment."
> >
> > [7] Chen et al. "LLMs are Highly-Constrained Biophysical Sequence Optimizers."
> >
> > [8] Stanton et al. "Closed-Form Test Functions for Biophysical Sequence Optimization Algorithms."
> >
> > [9] De Visser, Cooper, and Elena. "The causes of epistasis."

---

> ### Author Response · Authors · 2024-11-25
> **Any follow up questions**
>
> Dear Reviewer QrUQ,
>
> Given that the end of the discussion period is approaching, we would like to ask if you have any further concerns or questions, particularly as a follow-up to our response?  If we have addressed all your concerns, we hope you will consider increasing your score.
>
> Thank you in advance!

---

> > ### Author Response · Authors · 2024-11-30
> > **Disscussion period ending**
> >
> > Dear Reviewer QrUQ,
> >
> > Given that the discussion period is ending, we would like to ask if there any additional questions or concerns you would like you to address?

---

> > > ### Comment · Reviewer_QrUQ · 2024-12-01
> > > **Thank you for the responses**
> > >
> > > I would like to thank the authors for their additional experiments and for providing more context on the protein modeling domain. Most of my concerns have been addressed, and I am raising my score accordingly.
> > >
> > >
> > > If I may suggest some additions to the current presentation of future work and particularly generalizing this architecture to other domains – while my original concerns with the method have mostly been addressed by the author responses, the responses focus on the protein domain, and arguably more empirical study is required (that could be outside the scope of this paper) to check whether concerns regarding global controllability via local edits and the train-test mismatch of the amount of masking show up when applying the proposed method to other domains such as natural language. I thus encourage the authors to add some discussion about these potential challenges when describing the future extension of the proposed method to other domains.

---

### Official Review · Reviewer_ey43 · 2024-11-10

**Soundness:** 3
**Presentation:** 3
**Contribution:** 3
**Rating:** 6
**Confidence:** 4

**Summary:**

The authors apply concept bottleneck language models to protein design. They propose this model has three benefits; control, interpretability, and facilitated debugging. To support this, they demonstrate evidence that the model can optimize for single or double property control, better than existing conditioning approaches. They showcase this by redesigning the hydrophobic patch of an antibody, showing that it responds to the in-silico metrics as expected. Finally, they correlate model weights with trained concepts, and propose these correlations can help model debugging efforts

**Strengths:**

I believe this CB-pLM is the first model which applies concept bottleneck models to protein design, the authors show with strong evidence that CB-pLM effectively shifts concept distributions better than existing approaches such as C-pLMs and CC-pLMs. The paper is well written, and does a good job at showcasing the in-silico results for conditioning.

**Weaknesses:**

A limitation of this approach is that biophysical concepts need to be explicitly defined for controlling design. The argument that a CB-pLM is more interpretable because of the training approach is less convincing, as the results appear to be evidence that the model learns the provided concepts correctly, and doesn’t expand to a new interpretation of known biological properties. Concepts used in the model are also straightforward to calculate and interpret the outputs of with bioinformatics tools, alone. The model does not provide additional interpretability on proteins that otherwise exists.  Similarly, while the redesign example appears to currently be tested in a lab, it is not possible to asses if the model improves the solubility and function of  Siltuximab, based on the data provided.

**Questions:**

* Line 245: it appears the time is denoted as a typo
* Table 1: how was the validation set chosen, is there overlap between the validation set and LBSTER/ESM2 train sets?
* Section 4.1
    * Why was the C-pLM baseline tested with random substitutions instead of feature attribution.
    * Model size does not appear to improve intervention accuracy or change, significantly.
* Section 4.2
    * What is the motivation for using WSJ here. Why not use ESM2 or an inverse-design approach like ProteinMPNN or ESM-IF, for sequence redesign in the hydrophobic region?
    * It seems useful to include a non deep-learning baseline here that at minimum randomly re-samples residues in the hydrophobic patch with hydrophilic residues.
    * Is the hydrophobic patch critical for the function of the antibody? How do you know if the redesigned region has improved or reduced function.
* Section 5:
     * Is it not expected that the model should correctly learn or not learn the concepts provided during training?
     * It is also clear in what application the model debugging of a CB-pLM might be relevant, based on the given example.

---

> ### Author Response · Authors · 2024-11-22
> **Response to Reviewer ey43 part 1**
>
> Thank you for the detailed review and feedback. First, we address the weaknesses and then respond to your questions point-by-point.
>
> ## Re: Weaknesses
>
> - **Concepts need to be explicitly defined for controlling design.**
>   To understand and control the concepts used by a model for generation, one can either explicitly add these concepts to the model, as we do in our paper, or use post-hoc methods (e.g., probes, SAEs, PCA) to identify them. Both approaches have tradeoffs. Our method requires defining concepts and some data annotation beforehand, while post-hoc methods involve a significant effort to analyze the model's representations after training. Post-hoc methods face challenges such as explanation faithfulness and concept disentanglement. These methods might capture elements in the model’s representations that the model doesn't actually use to determine its output. Even if a concept can be predicted from the representations, it may not be important to the model. Moreover, identifying a concept in the representations doesn't ensure reliable control over it, as the model may represent that concept alongside other correlated factors---disentanglement. By explicitly defining the concepts we want to control ahead of time, we avoid these issues. This approach requires more upfront work during training.
>   Please note that we do not need the entire dataset to be fully annotated; a subset is often sufficient to learn the desired concepts. For example, when training, we add all concepts in SWISS-PROT; however, UniRef50 data do not have these concepts annotated in them, so we only have less than 3% of the data fully annotated.
>
> - **The argument that a CB-pLM is more interpretable is not convincing. CB-pLM doesn’t expand to a new interpretation of known biological properties.**
>   There are two types of interpretability offered by CB-pLM:  **Local Interpretability:**   This means that when generating a sequence, the model can:
>     a) Indicate which concepts it relies on for its predictions.
>     b) Accurately identify the input tokens that influence its output.
>     We achieve the first point through our concept-bottleneck module and associated losses, and the second point through our proposed feature masking and regulation scheme.
>  **Global Interpretability:**   This involves insights into the relationship learned by the model between predefined concepts and various tokens. This is achieved through the linear decoder; for instance, our model can localize which amino acids are crucial for protein hydrophobicity. We have clarified this point in section 5.1 of the updated manuscript. Interpretations beyond the predefined concepts are outside the scope of this work.
>
> - **It is not possible to assess if the model improves the solubility and function of Siltuximab, based on the data provided.**
>   Please find our response in the general response **R2**.

---

> > ### Author Response · Authors · 2024-11-22
> > **Response to Reviewer ey43 part 2**
> >
> > ### Re: Questions
> >
> > - **Line 245: typo**
> >   Thank you for catching this. We have corrected this in the updated manuscript.
> >
> > - **Table 1: Overlap between validation dataset and open source model’s training dataset**
> >   You are correct; there might have been an overlap between our validation dataset and the open-source models training dataset. Thank you for catching this; we repeated the experiment but replaced the uniref50/SWISS-PORT validation dataset with 10K randomly sampled antibodies from the publicly available Mason dataset [1]. No models have reported training on this dataset, so we assume it is safe to use it for a fair comparison. We have updated manuscript section 3.1 with the new results. Overall, we find that when comparing to models that do not require extra information during inference (i.e., no additional tags), our models are on par with open-source models of similar size. Notably, our model always produces better results than the ESM2 model with a similar size, suggesting that forcing the model to learn bio-physical concepts might result in an overall better model.
> >
> > - **Section 4.1: C-pLM baseline tested with random substitutions instead of feature attribution**
> >   In C-pLM, there is no concept classifier, i.e., the representations of the model are inscrutable, so one cannot know which part of the representation encodes the concept of interest. To identify the tokens that influence a concept, one would backpropagate from a concept prediction to the input tokens; there is no way to do this in this setting.
> >
> > - **Section 4.1: Model size does not appear to improve intervention accuracy significantly**
> >   Please find our response in the general response **R1**.
> >
> > - **Section 4.2: Motivation for using WJS here over ESM2 or an inverse-design approach**
> >   WJS is more recent than ESM2, and in benchmarking their approach against ESM2, the WJS paper demonstrated superior performance in this task. Notably, the WJS paper received an Outstanding Paper Award at ICLR2024. Therefore, we focused on WJS as an example of an unguided generation model. However, per your request, we have also added ESM2 as a baseline. Please refer to Figures 7a, 7b, 20, 21, and 22 in the updated manuscript for the results. As expected, the designs produced by WJS were superior to those by ESM2, especially in terms of naturalness (Figure 20). Regarding IF methods, they have been trained on PDB data and have been shown to be unreliable for antibodies [7,8]; therefore, we excluded them from this experiment.
> >
> > - **Section 4.2: Non-deep learning baseline randomly re-samples residues in the hydrophobic patch with hydrophilic residues**
> >   As per your request, we have that as a baseline. Results are found in Figures 7a, 7b, 20, 21, and 22. This baseline is denoted as Hydrophilic Resample. We found that these designs tend to negatively impact the naturalness of the protein (Figure 20).
> >
> > - **Section 4.2: Is the hydrophobic patch critical for the function of the antibody? How do you know if the redesigned region has improved or reduced function?**
> >   Hydrophobic patches are crucial for antibody function as they contribute to molecular stability and enable essential interactions with biological targets. Notably, siltuximab, which is currently approved for medical use, must be administered intravenously due to poor stability—a less preferred method compared to subcutaneous delivery. Improving stability properties could allow for subcutaneous delivery, enhancing patient compliance by reducing hospital visits for infusions. The goal is to propose designs that reduce aggregation at high concentrations by modulating hydrophobicity, without compromising the antibody's functionality, which is measured by its binding affinity to a given target.
> >
> > - **Section 5: Is it not expected that the model should correctly learn or not learn the concepts provided during training?**
> >   Yes, that is our goal. In scientific applications of machine learning, domain experts often bring prior knowledge and want to know whether a model’s representations encode certain biophysical attributes. Additionally, they may wish to modulate or control these attributes. Determining whether a model’s representations encode a specific property is challenging [9,10,11] and there is often no guarantee that a high-performing model will learn the features a domain expert wants to control. Our proposal addresses both of these issues. The concept bottleneck layer acts as a mechanism to enable understanding and control of factors that are important to domain experts.
> >
> > - **Section 5: It is also clear in what application the model debugging of a CB-pLM might be relevant, based on the given example.**
> >   Please find our response in the general response **R3**.

---

> > > ### Author Response · Authors · 2024-11-22
> > > **Response to Reviewer ey43 part 3**
> > >
> > > ### References
> > >
> > > [1] Mason, D. M., et al. "Optimization of therapeutic antibodies by predicting antigen specificity from antibody sequence via deep learning."
> > >
> > > [2] Li, Francesca-Zhoufan, et al. "Feature reuse and scaling: Understanding transfer learning with protein language models."
> > >
> > > [3] Frey, Nathan C., et al. "Cramming Protein Language Model Training in 24 GPU Hours."
> > >
> > > [4] Fournier, Quentin, et al. "Protein Language Models: Is Scaling Necessary?"
> > >
> > > [5] Elnaggar, Ahmed, et al. "Ankh: Optimized protein language model unlocks general-purpose modelling."
> > >
> > > [6] D’Amour, Alexander, et al. "Underspecification presents challenges for credibility in modern machine learning."
> > >
> > > [7] Brennan Abanades, et al. "ImmuneBuilder: Deep-Learning models for predicting the structures of immune proteins."
> > >
> > > [8] Kenlay, Henry, et al. "ABodyBuilder3: Improved and scalable antibody structure predictions."
> > >
> > > [9] Hewitt, John, and Percy Liang. "Designing and interpreting probes with control tasks."
> > >
> > > [10] Kumar, Abhinav, Chenhao Tan, and Amit Sharma. "Probing classifiers are unreliable for concept removal and detection."
> > >
> > > [11] D’Amour, Alexander, et al. "Underspecification presents challenges for credibility in modern machine learning."

---

> ### Author Response · Authors · 2024-11-25
> **Any follow up question?**
>
> Dear Reviewer ey43,
>
>  Given that the end of the discussion period is approaching, we would like to ask if you have any further concerns or questions, particularly as a follow-up to our response?  If we have addressed all your concerns, we hope you will consider increasing your score.
>
> Thank you in advance!

---

> ### Comment · Reviewer_ey43 · 2024-11-25
>
> Thank you for addressing my questions. Specifically, I regrettably missed that only 3% of the data was annotated during training and can appreciate that it is quite impressive that the model can learn the correct and translatable concepts with so few annotations.  I also appreciate the thoughtful response to the rest of my questions. I still am not convinced that the CB framework helps with model debugging. However, after reading all the reviews and your responses I have increased my score accordingly and have no follow up questions.

---

### Author Response · Authors · 2024-11-22
**General response part 1**

We would like to thank all the reviewers for their valuable feedback and for noting that the paper is well written (Reviewers ey43, QrUQ, i9uk, AGed), adapting concept bottleneck model to MLM is novel (Reviewer QrUQ), the use of integrated gradients to find tokens to resample is novel (Reviewer QrUQ), the idea is well executed and leads to novel applications (Reviewer i9uk), and that our approach is compelling (Reviewer AGed).

_________
## General Response

-   **Scaling the model size does not seem to improve performance beyond a certain size (Reviewers ey43, QrUQ, i9uk)**

    -  **R1:** This observation is consistent with existing literature. Numerous studies [1,2,3,4] have found that scaling protein language models beyond a certain point does not result in significant performance gains. We speculate that this may be due to overparameterization; larger models require more training data to effectively utilize the additional parameters, and the amount of data available for proteins is considerably less than that for natural language. Additionally, the Rashomon set literature [5] suggests that there is typically no correlation between model size and the steerability or interpretability of the model.

-   **No wet lab validations (Reviewers ey43, AGed)**

    -   **R2:** We agree that wet-lab results would significantly bolster confidence in our proposed method. However, it is crucial to understand that completely verifying whether a redesigned antibody has improved developability while retaining potency is a complex and time-consuming process that can span months or even years. ***Our preliminary findings are promising; in one experiment, we observed that 46% of antibodies redesigned using CB-pLM successfully bound to the original target antigen, and 15% even demonstrated improved binding over the original antibody. These results strongly indicate that CB-pLM is capable of producing functional proteins.***
        Moreover, our preliminary wet-lab experiments revealed a high inverse correlation between hydrophobicity in the Therapeutic Antibody Profiler (TAP) and the structural aggregation propensity (SAP) metrics with the molecule's Hydrophobic Interaction Chromatography (HIC) conductivity, which is currently being measured in the wet lab. This observed inverse correlation motivated us to perform additional experiments calculating SAP/TAP metrics (physics-based computations) of folded designs while ensuring they maintain the naturalness of the seed. We plotted these hydrophobicity SAP/TAP metrics in Figure 21 of the updated manuscript and expect designs in the lower left quadrant to reduce hydrophobicity compared to the seed, as observed in previous wet-lab experiments. For further details, please refer to Appendix C.2.

        It is also important to emphasize that for the purposes of this paper, it is sufficient for us to demonstrate that the algorithm successfully manipulates the concepts selected by the user. We have achieved this for many concepts that are less costly to evaluate. While we continue to pursue further experimental validation, it is worth noting that numerous other studies [6,7,8,9,10] have already demonstrated that in-silico results in protein design can indeed be translated to wet-lab results.

-   **Model debugging (Reviewers ey43, e4sF)**

    -   **R3:** Model debugging refers to the ability to detect and fix a model's errors, either during training or at test time. In the standard machine learning pipeline, it can be difficult to ascertain the root cause of a model's error, such as issues with optimization, initialization, training data, or function class. The current paradigm often relies on painstaking manual effort or retraining (or fine-tuning) to address errors. Our proposal introduces new mechanisms to an MLM that allow us to attribute certain model errors to human-understandable concepts. Critically, this enables us to answer the following questions:

        1. Did my model learn an important feature of interest? Currently, the prevailing alternative is to train probes (or SAEs) on model representations to determine whether a feature of interest can be predicted. However, consistent results [11,12,13] show that just because a model's representation is correlated with--or predictive of--a feature does not mean the model bases its output on that feature.

        2. Is my model relying on a spurious feature, according to the domain expert, as the basis for generating certain outputs? In which situations is the model likely to fail? Inspecting the weights of the linear layer can answer such questions.

        3. How can I intervene to correct unwanted correlations? The concept bottleneck (CB) layer allows for simple interventions to correct any unwanted correlations.

        By incorporating these capabilities, our approach provides a more effective and transparent way to debug and improve models in scientific applications. We have clarified this in section 5.2.

---

> ### Author Response · Authors · 2024-11-22
> **General response part 2**
>
> ## Changes in Manuscript
> We would like to highlight the following changes to the manuscript to address reviewers' concerns:
> - We have repeated the perplexity experiment in section 3.1 using a publicly available antibodies dataset Mason  [14] as a test dataset to ensure that there is no cross-contamination between the test dataset and the training dataset of open-source models (Reviewer ey43).
> - Added ESM2-650M and ESM2-3B as baselines in section 3.1 (Reviewer e4sF).
> - Added ESM2 and  Hydrophilic Resample as baselines for the Siltuximab experiment in section 4.2 (Reviewer ey43).
> - Added TAP metrics as an additional validation for protein naturalness (Reviewer QrUQ).
> - Added Hydrophobicity SAP/TAP metrics for Siltuximab redesigns as an additional proxy to wet-lab results (Reviewers ey43, AGed).
> - Repeated the control experiments with 25% masking (Reviewer QrUQ).
> _________
> ### References
>
> [1] Li, Francesca-Zhoufan, et al. "Feature reuse and scaling: Understanding transfer learning with protein language models."
>
> [2] Frey, Nathan C., et al. "Cramming Protein Language Model Training in 24 GPU Hours."
>
> [3] Fournier, Quentin, et al. "Protein Language Models: Is Scaling Necessary?"
>
> [4] Elnaggar, Ahmed, et al. "Ankh: Optimized protein language model unlocks general-purpose modelling."
>
> [5] D'Amour, Alexander, et al. "Underspecification presents challenges for credibility in modern machine learning, 2020."
>
> [6] Stanton, Samuel, et al. "Accelerating bayesian optimization for biological sequence design with denoising autoencoders." International Conference on Machine Learning.
>
> [7] Gruver, Nate, et al. "Protein design with guided discrete diffusion."
>
> [8] Frey, Nathan C., et al. "Protein discovery with discrete walk-jump sampling."
>
> [9] Alamdari, S., et al. "Protein generation with evolutionary diffusion: Sequence is all you need."
>
> [10] Tagasovska, Nataša, et al. "Implicitly Guided Design with PropEn: Match your Data to Follow the Gradient."
>
> [11] Hewitt and Liang. "Designing and interpreting probes with control tasks."
>
> [12] Kumar, Tan, and Sharma. "Probing classifiers are unreliable for concept removal and detection."
>
> [13] D’Amour et al. "Underspecification presents challenges for credibility in modern machine learning."
>
> [14] Mason, D. M., et al. "Optimization of therapeutic antibodies by predicting antigen specificity from antibody sequence via deep learning."

---

### Meta-Review · Area_Chair_kzjY · 2024-12-18

**Metareview:**

This paper introduces "Concept Bottleneck Protein Language Models" (CB-pLM), a novel approach that integrates a concept bottleneck layer into protein language models. This architecture aims to enhance control, interpretability, and debugging capabilities without compromising performance. The authors demonstrate that CB-pLM can effectively manipulate protein properties, offering significant improvements in controllability and interpretability over existing models, and that they can be scaled to the billion-parameter-sized regime.

The main strengths of this paper are:
* Innovative Architecture: The integration of a concept bottleneck layer into protein language models is novel and offers significant improvements in control and interpretability.
* Empirical Validation of Controllability: The paper provides strong empirical evidence that CB-pLM can manipulate protein properties more effectively than existing models.
* Scalability: The model is shown to scale up to 3 billion parameters, demonstrating its robustness and potential for large-scale applications.
* Practical Applications: The ability to control and interpret protein properties has significant implications for drug discovery and other bio applications.

The main weaknesses, in turn, are:
* Need for concept specification: a typical downside of concept bottleneck layers that applies to CB-pLM too is that the "concepts" need to be explicitly specified. This can be done either with prior knowledge or, as suggested by the authors, using a post-hoc interpretability method.
* Limited/preliminary experimental results for "debugging": multiple reviewers raised concerns about the replicability/significance of this part of the experimental validation.
* Limited applicability and generalizability of the approach beyond protein sequences.

All things considered, this paper provides a modest but compelling contribution on an important open research problem, namely interpretability in generative AI for biology, which is of high potential impact. Hence, I recommend acceptance.

**Additional Comments On Reviewer Discussion:**

The main points raised by the reviewers were:

* Reviewer ey43 raised concerns about the scalability of the method and the lack of wet-lab validation. They also expressed skepticism about the novelty of the insights derived from this model, pointing out at that a similar level of interpretation is achievable with existing standard bioinformatics tools. The authors responded by acknowledging the complexity and time required for wet lab experiments and provided preliminary wet-lab validation results to support their claims. The reviewer appreciated the detailed responses and increased their score, but maintained skepticism about the usefulness of the method for debugging.
* Reviewer QrUQ questioned the general applicability of the model to other domains and expressed concerns about the naturalness of generated sequences not being explicitly enforced during generation. The authors provided additional experiments demonstrating naturalness preservation and clarified the domain-specific requirements, which addressed most of the reviewer’s concerns, leading to an increased score.
* Reviewer e4sF highlighted issues with the writing, clarity and focus of the manuscript and the need for more empirical validation. In particular, this reviewer requested precise definition of certain notions (eg "human-understandable"). The authors revised the manuscript to address these concerns and provided additional comparisons with state-of-the-art models, resulting in a higher score from the reviewer.
* Reviewer i9uk found the controllability experiments convincing (and a sufficient contribution) but not the controllability ones, which they found to be cursory, subjectively evaluated and not entirely reproducible. The reviewer also suggested further discussion on comparison to other models and asked whether the proposed method was given additional information that others werent. The authors clarified the comparison methodology and corrected minor typos. The reviewer did not engage with the response.
* Reviewer AGed emphasized the importance of wet lab validation and biosecurity risks. The authors acknowledged these points and included a discussion on biosecurity risks in the revised manuscript.

In summary, the paper was initial met with very mixed reviews, most of which praised the novelty of the method and importance of the problem, but expressed concerns about the soundness, significance, replicability, and objectivity of at least some of the results. However, after a remarkably thorough and compelling rebuttal, 3 reviewers increased their score, bringing the paper to an almost unanimous consensus for (weak) acceptance.

---

### Decision · Program_Chairs · 2025-01-22

Accept (Poster)